# Boosting First-Order Methods by Shifting Objective: New Schemes with Faster Worst-Case Rates

**Kaiwen Zhou**[†]          **Anthony Man-Cho So**[‡]          **James Cheng**[†]
[†]Department of Computer Science and Engineering
[‡]Department of Systems Engineering and Engineering Management
The Chinese University of Hong Kong
kwzhou@cse.cuhk.edu.hk; manchoso@se.cuhk.edu.hk; jcheng@cse.cuhk.edu.hk

## Abstract

We propose a new methodology to design first-order methods for unconstrained strongly convex problems. Specifically, instead of tackling the original objective directly, we construct a shifted objective function that has the same minimizer as the original objective and encodes both the smoothness and strong convexity of the original objective in an interpolation condition. We then propose an algorithmic template for tackling the shifted objective, which can exploit such a condition. Following this template, we derive several new accelerated schemes for problems that are equipped with various first-order oracles and show that the interpolation condition allows us to vastly simplify and tighten the analysis of the derived methods. In particular, all the derived methods have faster worst-case convergence rates than their existing counterparts. Experiments on machine learning tasks are conducted to evaluate the new methods.

## 1 Introduction

In this paper, we focus on the following unconstrained smooth strongly convex problem:

$$\min_{x \in \mathbb{R}^d} f(x) = \frac{1}{n} \sum_{i=1}^{n} f_i(x), \tag{1}$$

where each $f_i$ is $L$-smooth and $\mu$-strongly convex,[1] and we denote $x^\star \in \mathbb{R}^d$ as the solution of this problem. The $n = 1$ case covers a large family of classic strongly convex problems, for which gradient descent (GD) and Nesterov's accelerated gradient (NAG) [32, 33, 35] are the methods of choice. The $n \geq 1$ case is the popular finite-sum case, where many elegant methods that incorporate the idea of variance reduction have been proposed. Problems with a finite-sum structure arise frequently in machine learning and statistics, such as empirical risk minimization (ERM).

In this work, we tackle problem (1) from a new angle. Instead of designing methods to solve the original objective function $f$, we propose methods that are designed to solve a shifted objective $h$: $\min_{x \in \mathbb{R}^d} h(x) = \frac{1}{n} \sum_{i=1}^{n} h_i(x)$, where $h_i(x) = f_i(x) - f_i(x^\star) - \langle \nabla f_i(x^\star), x - x^\star \rangle - \frac{\mu}{2}\|x - x^\star\|^2$. It can be easily verified that each $h_i(x)$ is $(L-\mu)$-smooth and convex, $\nabla h_i(x) = \nabla f_i(x) - \nabla f_i(x^\star) - \mu(x - x^\star)$, $\nabla h(x) = \nabla f(x) - \mu(x - x^\star)$, $h_i(x^\star) = h(x^\star) = 0$ and $\nabla h_i(x^\star) = \nabla h(x^\star) = \mathbf{0}$, which means that the shifted problem and problem (1) share the same optimal solution $x^\star$. Let us write a well-known property of $h$:

$$\forall x, y \in \mathbb{R}^d, h(x) - h(y) - \langle \nabla h(y), x - y \rangle \geq \frac{1}{2(L-\mu)}\|\nabla h(x) - \nabla h(y)\|^2, \tag{2}$$

which encodes both the smoothness and strong convexity of $f$. The discrete version of this inequality is equivalent to the *smooth strongly convex interpolation condition* discovered in [51]. As studied in [51], this type of inequality forms a necessary and sufficient condition for the existence of a smooth strongly convex $f$ interpolating a given set of triples $\{(x_i, \nabla f_i, f_i)\}$, while the usual collection of $L$-smoothness and strong convexity inequalities is only a necessary condition.[2] For worst-case analysis, it implies that tighter results can be derived by exploiting condition (2) than using smoothness and strong convexity "separately", which is common in existing worst-case analysis. We show that our methodology effectively exploits this condition and consequently, we propose several methods that achieve faster worst-case convergence rates than their existing counterparts.

In summary, our methodology and proposed methods have the following distinctive features:

- We show that our methodology works for problems equipped with various first-order oracles: deterministic gradient oracle, incremental gradient oracle and incremental proximal point oracle.

- We leverage a cleaner version of the interpolation condition discovered in [51], which leads to simpler and tighter analysis to the proposed methods than their existing counterparts.

- For our proposed stochastic methods, we deal with shifted variance bounds / shifted stochastic gradient norm bounds, which are different from all previous works.

- All the proposed methods achieve faster worst-case convergence rates than their counterparts that were designed to solve the original objective $f$.

Our work is motivated by a recently proposed robust momentum method [10], which converges under a Lyapunov function that contains a term $h(x) - \frac{1}{2(L-\mu)}\|\nabla h(x)\|^2$. Our work conducts a comprehensive study of the special structure of this term.

This paper is organized as follows: In Section 2, we present high-level ideas and lemmas that are the core building blocks of our methodology. In Section 3, we propose an accelerated method for the $n = 1$ case. In Section 4, we propose accelerated stochastic variance-reduced methods for the $n \geq 1$ case with incremental gradient oracle. In Section 5, we propose an accelerated method for the $n \geq 1$ case with incremental proximal point oracle. In Section 6, we provide experimental results.

## 1.1 Notations and Definitions

In this paper, we consider problems in the standard Euclidean space denoted by $\mathbb{R}^d$. We use $\langle \cdot, \cdot \rangle$ and $\|\cdot\|$ to denote the inner product and the Euclidean norm, respectively. We let $[n]$ denote the set $\{1, 2, \ldots, n\}$, $\mathbb{E}$ denote the total expectation and $\mathbb{E}_{i_k}$ denote the conditional expectation given the information up to iteration $k$. We say that a convex function $f : \mathbb{R}^d \to \mathbb{R}$ is *L-smooth* if it has $L$-Lipschitz continuous gradients, i.e., $\forall x, y \in \mathbb{R}^d, \|\nabla f(x) - \nabla f(y)\| \leq L\|x - y\|$. Some important consequences of this assumption can be found in the textbook [35]: $\forall x, y \in \mathbb{R}^d$, $\frac{1}{2L}\|\nabla f(x) - \nabla f(y)\|^2 \leq f(x) - f(y) - \langle \nabla f(y), x - y \rangle \leq \frac{L}{2}\|x - y\|^2$. We refer to the first inequality as *interpolation condition* following [51]. A continuously differentiable $f$ is called *$\mu$-strongly convex* if $\forall x, y \in \mathbb{R}^d, f(x) - f(y) - \langle \nabla f(y), x - y \rangle \geq \frac{\mu}{2}\|x - y\|^2$. Given a point $x \in \mathbb{R}^d$, an index $i \in [n]$ and $\alpha > 0$, a deterministic oracle returns $(f(x), \nabla f(x))$, an incremental first-order oracle returns $(f_i(x), \nabla f_i(x))$ and an incremental proximal point oracle returns $(f_i(x), \nabla f_i(x), \text{prox}_i^\alpha(x))$, where the proximal operator is defined as $\text{prox}_i^\alpha(z) = \arg\min_x \{f_i(x) + \frac{\alpha}{2}\|x - z\|^2\}$. We denote $\epsilon > 0$ as the required accuracy for solving problem (1) (i.e., to achieve $\|x - x^\star\|^2 \leq \epsilon$), which is assumed to be small. We denote $\kappa \triangleq L/\mu$, which is often called the condition ratio.

## 1.2 Related Work

Problem (1) with $n = 1$ is the classic smooth strongly convex setting. Standard analysis shows that for this problem, GD with $\frac{2}{L+\mu}$ stepsize converges linearly at a $\left(\frac{\kappa-1}{\kappa+1}\right)^2$ rate[3] (see the textbook [35]). The heavy-ball method [40] fails to converge globally on this problem [28]. The celebrated NAG is proven to achieve a faster $1 - 1/\sqrt{\kappa}$ rate [35]. This rate remains the fastest one until recently, Van Scoy et al. [53] proposed the Triple Momentum method (TM) that converges at a $(1 - 1/\sqrt{\kappa})^2$

rate. Numerical results in [29] suggest that this rate is not improvable. In terms of reducing $\|x - x^\star\|^2$ to $\epsilon$, TM is stated to have an $O\big((\sqrt{\kappa}/2)(\log \frac{1}{\epsilon} + \log \sqrt{\kappa})\big)$ iteration complexity (cf. Table 2, [53]) compared with the $O(\sqrt{\kappa} \log \frac{1}{\epsilon})$ complexity of NAG. In the general convex setting, recent works [21, 3, 23] propose new schemes that have lower complexity than the original NAG. Several of these new schemes were discovered based on the recent works that use semidefinite programming to study worst-case performances of first-order methods. Starting from the performance estimation framework introduced in [14], many different approaches and extensions have been proposed [28, 48, 51, 50, 49].

For the $n \geq 1$ case, stochastic gradient descent (SGD) [41], which uses component gradients $\nabla f_i(x)$ to estimate the full gradient $\nabla f(x)$, achieves a lower iteration cost than GD. However, SGD only converges at a sub-linear rate. To fix this issue, various variance reduction techniques have been proposed recently, such as SAG [43, 44], SVRG [20, 56], SAGA [12], SDCA [46] and SARAH [36]. Inspired by the Nesterov's acceleration technique, accelerated stochastic variance-reduced methods have been proposed in pursuit of the lower bound $O(n + \sqrt{n\kappa} \log \frac{1}{\epsilon})$ [55], such as Acc-Prox-SVRG [37], APCG [31], ASDCA [47], APPA [15], Catalyst [30], SPDC [57], RPDG [27], Point-SAGA [11] and Katyusha [1]. Among these methods, Katyusha and Point-SAGA, representing the first two directly accelerated incremental methods, achieve the fastest rates. Point-SAGA leverages a more powerful incremental proximal operator oracle. Katyusha introduces the idea of negative momentum, which serves as a variance reducer that further reduces the variance of the SVRG estimator. This construction motivates several new accelerated methods [60, 2, 26, 24, 58, 59].

## 2  Tackling the Shifted Objective

As mentioned in the introduction, our methodology is to minimize the shifted objective[4] $h$ with the aim of exploiting the interpolation condition. However, a critical issue is that we cannot even compute its gradient $\nabla h(x)$ (or $\nabla h_i(x)$), which requires the knowledge of $x^\star$. We figured out that in some simple cases, a change of "perspective" is enough to access this gradient information. Take GD $x_{k+1} = x_k - \eta \nabla f(x_k)$ as an example. Based on the definition $\nabla h(x_k) = \nabla f(x_k) - \mu(x_k - x^\star)$, we can rewrite the GD update as $x_{k+1} - x^\star = (1 - \eta\mu)(x_k - x^\star) - \eta \nabla h(x_k)$, and thus

$$\|x_{k+1} - x^\star\|^2 = (1 - \eta\mu)^2 \|x_k - x^\star\|^2 \underbrace{-2\eta(1 - \eta\mu)\langle \nabla h(x_k), x_k - x^\star \rangle + \eta^2 \|\nabla h(x_k)\|^2}_{R_0}.$$

If we set $\eta = 2/(L+\mu)$, using the interpolation condition (2), we can conclude that $R_0 \leq 0$, which leads to a convergence guarantee. It turns out that this argument is just the one-line proof of GD in the textbook (Theorem 2.1.15, [35]) but looks more structured in our opinion. However, this change of "perspective" is too abstract for more complicated schemes. Our solution is to first fix a template updating rule, and then encode this idea into a technical lemma, which serves as an instantiation of the shifted gradient oracle. To facilitate its usage, we formulate this lemma with a classic inequality whose usage has been well-studied. Proofs in this section are given in Appendix A.

**Lemma 1** (Shifted mirror descent lemma). *Given a gradient estimator $\mathcal{G}_y$, vectors $z^+, z^-, y \in \mathbb{R}^d$, fix the updating rule $z^+ = \arg\min_x \big\{ \langle \mathcal{G}_y, x \rangle + \alpha/2 \|x - z^-\|^2 + \mu/2 \|x - y\|^2 \big\}$. Suppose that we have a shifted gradient estimator $\mathcal{H}_y$ satisfying the relation $\mathcal{H}_y = \mathcal{G}_y - \mu(y - x^\star)$, it holds that $\langle \mathcal{H}_y, z^- - x^\star \rangle = \frac{\alpha}{2}\big(\|z^- - x^\star\|^2 - (1 + \frac{\mu}{\alpha})^2\|z^+ - x^\star\|^2\big) + \frac{1}{2\alpha}\|\mathcal{H}_y\|^2.$*

**Remark 1.** *In general convex optimization, a similar lemma (for $\mathcal{G}$) serves as the core lemma for mirror descent[5] (e.g., Theorem 5.3.1 in the textbook [8]). This type of lemma also appears frequently in online optimization, which is used as an upper bound on the regret at the current iteration (e.g., Lemma 3 in [45]). In the strongly convex setting, unlike the common $(1 + \frac{\mu}{\alpha})^{-1}$ (or $1 - \frac{\mu}{\alpha}$) contraction ratio in existing work (e.g., Lemma 2.5 in [1]), Lemma 1 provides a $(1 + \frac{\mu}{\alpha})^{-2}$ ratio, which is one of the keys to the improved worst-case rates achieved in this paper.*

Lemma 1 allows us to choose various gradient estimators for $h$ directly, given that the relation $\mathcal{H}_x = \mathcal{G}_x - \mu(x - x^\star)$ holds for some practical $\mathcal{G}_x$. Here we provide some examples:

- Deterministic gradient: $\mathcal{H}_x^{\text{GD}} = \nabla h(x) \Rightarrow \mathcal{G}_x^{\text{GD}} = \nabla f(x)$.

- SVRG estimator: $\mathcal{H}_x^{\mathrm{SVRG}} = \nabla h_i(x) - \nabla h_i(\tilde{x}) + \nabla h(\tilde{x}) \Rightarrow \mathcal{G}_x^{\mathrm{SVRG}} = \nabla f_i(x) - \nabla f_i(\tilde{x}) + \nabla f(\tilde{x})$.
- SAGA estimator: $\mathcal{H}_x^{\mathrm{SAGA}} = \nabla h_i(x) - \nabla h_i(\phi_i) + \frac{1}{n}\sum_{j=1}^{n}\nabla h_j(\phi_j) \Rightarrow$
$$\mathcal{G}_x^{\mathrm{SAGA}} = \nabla f_i(x) - \nabla f_i(\phi_i) + \frac{1}{n}\sum_{j=1}^{n}\nabla f_j(\phi_j) - \mu\big(\frac{1}{n}\sum_{j=1}^{n}\phi_j - \phi_i\big).$$
- SARAH estimator: $\mathcal{H}_{x_k}^{\mathrm{SARAH}} = \nabla h_{i_k}(x_k) - \nabla h_{i_k}(x_{k-1}) + \mathcal{H}_{x_{k-1}}^{\mathrm{SARAH}}$ and $\mathcal{H}_{x_0}^{\mathrm{SARAH}} = \nabla h(x_0) \Rightarrow$
$$\mathcal{G}_{x_k}^{\mathrm{SARAH}} = \nabla f_{i_k}(x_k) - \nabla f_{i_k}(x_{k-1}) + \mathcal{G}_{x_{k-1}}^{\mathrm{SARAH}} \text{ and } \mathcal{G}_{x_0}^{\mathrm{SARAH}} = \nabla f(x_0).$$

It can be verified that the relation $\mathcal{H}_x = \mathcal{G}_x - \mu(x - x^\star)$ holds in all these examples. Note that it is important to ensure that $\mathcal{G}_x$ is practical. For example, the shifted stochastic gradient estimator $\nabla h_i(x) = [\nabla f_i(x) - \nabla f_i(x^\star)] - \mu(x - x^\star)$ does not induce a practical $\mathcal{G}_x$.

We also apply the idea of changing "perspective" to proximal operator $\mathrm{prox}_i^\alpha$ as given below.

**Lemma 2** (Shifted firm non-expansiveness). *Given relations $z^+ = \mathrm{prox}_i^\alpha(z^-)$ and $y^+ = \mathrm{prox}_i^\alpha(y^-)$, it holds that $\frac{1}{\alpha^2}\big(1 + \frac{2(\alpha+\mu)}{L-\mu}\big)\|\nabla h_i(z^+) - \nabla h_i(y^+)\|^2 + (1 + \frac{\mu}{\alpha})^2 \|z^+ - y^+\|^2 \le \|z^- - y^-\|^2$.*

**Remark 2.** *Recall the definition of a firmly non-expansive operator $T$ (e.g., Definition 4.1 in the textbook [6]): $\forall x, y, \|Tx - Ty\|^2 + \|(\mathrm{Id} - T)x - (\mathrm{Id} - T)y\|^2 \le \|x - y\|^2$. Lemma 2 can be derived by choosing[6] $T = (1 + \frac{\mu}{\alpha}) \cdot \mathrm{prox}_i^\alpha$ and strengthening $\langle Tx - Ty, (\mathrm{Id} - T)x - (\mathrm{Id} - T)y \rangle \ge 0$ using the interpolation condition. A similar lemma has also been used in the analysis of the proximal point algorithm [42]. In our problem setting, Defazio [11] also strengthened firm non-expansiveness, which produces a $(1 + \frac{\mu}{\alpha})^{-1}$ contraction ratio instead of the above $(1 + \frac{\mu}{\alpha})^{-2}$ ratio created by shifting objective.*

Now we have all the building blocks to migrate existing schemes to tackle the shifted objective. To maximize the potential of our methodology, we focus on developing accelerated methods. We can also tighten the analysis of non-accelerated methods, which could lead to new algorithmic schemes.

## 3 Deterministic Objectives

We consider the objective function (1) with $n = 1$. To begin, we recap the guarantee of NAG to facilitate the comparison. The proof is given in Appendix F for completeness. At iteration $K-1$, NAG produces $f(x_K) - f(x^\star) + \frac{\mu}{2}\|z_K - x^\star\|^2 \le (1 - 1/\sqrt{\kappa})^K (f(x_0) - f(x^\star) + \frac{\mu}{2}\|z_0 - x^\star\|^2)$, where $x_0, z_0 \in \mathbb{R}^d$ are the initial guesses. Denote the initial constant as $C_0^{\mathrm{NAG}} \triangleq f(x_0) - f(x^\star) + \frac{\mu}{2}\|z_0 - x^\star\|^2$. This guarantee shows that in terms of reducing $\|x - x^\star\|^2$ to $\epsilon$, the sequences $\{x_k\}$ (due to $f(x_K) - f(x^\star) \ge \frac{\mu}{2}\|x_K - x^\star\|^2$) and $\{z_k\}$ have the same iteration complexity $\sqrt{\kappa} \log \frac{2C_0^{\mathrm{NAG}}}{\mu\epsilon}$.

### 3.1 Generalized Triple Momentum Method

We present the first application of our methodology in Algorithm 1, which can be regarded as a technical migration[7] of NAG to the shifted objective. It turns out that Algorithm 1, when tuned optimally, is equivalent to TM [53] (except for the first iteration). We thus name it as Generalized Triple Momentum method (G-TM). In comparison with TM, G-TM has the following advantages:

- *Refined convergence guarantee.* TM has the guarantee (Eq.(11) in [10] with $\rho = 1 - 1/\sqrt{\kappa}$):

$$\|z_K - x^\star\|^2 \le \left(1 - \frac{1}{\sqrt{\kappa}}\right)^{2(K-1)} \left(\|z_1 - x^\star\|^2 + \frac{L-\mu}{L\mu}\left(h(y_0) - \frac{1}{2(L-\mu)}\|\nabla h(y_0)\|^2\right)\right),$$

which has an initial state issue: its initial constant correlates with $z_1$, which is not an initial guess. It can be verified that the first iteration of TM is GD with a $1/\sqrt{L\mu}$ stepsize, which exceeds the $2/(L+\mu)$ limit, and thus we do not have $\|z_1 - x^\star\|^2 \le \|z_0 - x^\star\|^2$ in general. This issue is possibly the reason for the $\log \sqrt{\kappa}$ factor stated in [53]. G-TM resolves this issue and removes the log factor.

- *More extensible proof.* Our proof of G-TM is based on Lemma 1, which, as mentioned in Section 2, allows shifted stochastic gradients. In comparison, the analysis of TM starts with establishing an algebraic identity and it is unknown whether this identity holds in the stochastic case.

**Algorithm 1** Generalized Triple Momentum (G-TM)

---

**Input:** $\{\alpha_k > 0\}, \{\tau_k^x \in ]0, 1[\}, \{\tau_k^z > 0\}$, initial guesses $y_{-1}, z_0 \in \mathbb{R}^d$ and iteration number $K$.

1: **for** $k = 0, \ldots, K - 1$ **do**

2:      $y_k = \tau_k^x z_k + (1 - \tau_k^x) y_{k-1} + \tau_k^z \big( \mu(y_{k-1} - z_k) - \nabla f(y_{k-1}) \big)$.

3:      $z_{k+1} = \arg\min_x \Big\{ \langle \nabla f(y_k), x \rangle + (\alpha_k/2) \|x - z_k\|^2 + (\mu/2) \|x - y_k\|^2 \Big\}$.

4: **end for**

**Output:** $z_K$.

---

- *General scheme.* The framework of G-TM covers both NAG and TM (Appendix B.1). When $\mu = 0$, it also covers the optimized gradient method [21], which is discussed in Section 7.

A subtlety of Algorithm 1 is that it requires storing a past gradient vector, and thus at the first iteration, two gradient computations are needed. The analysis of G-TM is based on the same Lyapunov function in [10]: $T_k = h(y_{k-1}) - \frac{1}{2(L-\mu)} \|\nabla h(y_{k-1})\|^2 + \frac{\lambda}{2} \|z_k - x^\star\|^2$, where $\lambda > 0$. In the following theorem, we establish the per-iteration contraction of G-TM and the proof is given in Appendix B.2.

**Theorem 1.** *In Algorithm 1, if we fix $\tau_k^z = \frac{1 - \tau_k^x}{L - \mu}, \forall k$ and choose $\{\alpha_k\}, \{\tau_k^x\}$ under the constraints $2\alpha_k \geq L\tau_k^x - \mu$ and $(1 + \frac{\mu}{\alpha_k})^2 (1 - \tau_k^x) \leq 1$, the iterations satisfy the contraction $T_{k+1} \leq (1 + \frac{\mu}{\alpha_k})^{-2} T_k$ with $\lambda = \frac{(\tau_k^x - \mu\tau_k^z)(\alpha_k + \mu)^2}{\alpha_k}$.*

When the constraints hold as equality, we derive a simple constant choice for G-TM: $\alpha = \sqrt{L\mu} - \mu, \tau_x = \frac{2\sqrt{\kappa} - 1}{\kappa}, \tau_z = \frac{\sqrt{\kappa} - 1}{L(\sqrt{\kappa} + 1)}$. Here we also provide the parameter choices of NAG and TM under the framework of G-TM for comparison. Detailed derivation is given in Appendix B.1.

$$
\text{NAG} \begin{cases} \alpha = \sqrt{L\mu} - \mu; \\ \tau_k^x = (\sqrt{\kappa} + 1)^{-1}, \tau_k^z = 0, & k = 0; \\ \tau_k^x = (\sqrt{\kappa})^{-1}, \tau_k^z = \frac{1}{L + \sqrt{L\mu}}, & k \geq 1. \end{cases} \qquad \text{TM} \begin{cases} \alpha = \sqrt{L\mu} - \mu; \\ \tau_k^x = (\sqrt{\kappa} + 1)^{-1}, \tau_k^z = 0, & k = 0; \\ \tau_k^x = \frac{2\sqrt{\kappa} - 1}{\kappa}, \tau_k^z = \frac{\sqrt{\kappa} - 1}{L(\sqrt{\kappa} + 1)}, & k \geq 1. \end{cases}
$$

Using the constant choice in Theorem 1, telescoping the contraction from iteration $K - 1$ to 0, we get

$$
\frac{\mu}{2} \|z_K - x^\star\|^2 \leq \left( 1 - \frac{1}{\sqrt{\kappa}} \right)^{2K} \left( \frac{\kappa - 1}{2\kappa} \left( h(y_{-1}) - \frac{1}{2(L - \mu)} \|\nabla h(y_{-1})\|^2 \right) + \frac{\mu}{2} \|z_0 - x^\star\|^2 \right). \quad (3)
$$

Denoting the initial constant as $C_0^{\text{G-TM}} \triangleq \frac{\kappa - 1}{2\kappa} (h(y_{-1}) - \frac{1}{2(L-\mu)} \|\nabla h(y_{-1})\|^2) + \frac{\mu}{2} \|z_0 - x^\star\|^2$, if we align the initial guesses $y_{-1} = x_0$ with NAG, we have $C_0^{\text{G-TM}} \ll C_0^{\text{NAG}}$. This guarantee yields a $\frac{\sqrt{\kappa}}{2} \log \frac{2C_0^{\text{G-TM}}}{\mu\epsilon}$ iteration complexity for G-TM, which is at least two times lower than that of NAG and does not suffer from an additional $\log \sqrt{\kappa}$ factor as is the case for the original TM.

### 3.1.1 The Tightness of (3)

It is natural to ask how tight the worst-case guarantee (3) is. We show that for the quadratic[8] $f(x) = \frac{1}{2} \langle D^\kappa x, x \rangle$ where $D^\kappa \triangleq \text{diag}(L, \mu)$ is a diagonal matrix, G-TM converges exactly at the rate in (3). Note that for this objective, $h(x) - \frac{1}{2(L-\mu)} \|\nabla h(x)\|^2 \equiv 0$, which means that the guarantee becomes $\|z_K - x^\star\|^2 \leq (1 - \frac{1}{\sqrt{\kappa}})^{2K} \|z_0 - x^\star\|^2$. Expanding the recursions in Algorithm 1, we obtain the following result and its proof is given in Appendix B.3.

**Proposition 1.1.** *If $f(x) = \frac{1}{2} \langle D^\kappa x, x \rangle$, G-TM produces $\|z_K - x^\star\|^2 = \left( 1 - \frac{1}{\sqrt{\kappa}} \right)^{2K} \|z_0 - x^\star\|^2$.*

## 4 Finite-Sum Objectives with Incremental First-Order Oracle

We now consider the finite-sum objective (1) with $n \geq 1$. We choose SVRG [20] as the base algorithm to implement our boosting technique, and we also show that an accelerated SAGA [12] variant can be similarly constructed in Section 4.2. Proofs in this section are given in Appendix C.

**Algorithm 2** SVRG Boosted by Shifting objective (BS-SVRG)

---

**Input:** Parameters $\alpha > 0, \tau_x \in ]0,1[$, initial guess $x_0 \in \mathbb{R}^d$, epoch number $S$ and epoch length $m$.

**Initialize:** Vectors $z_0^0 = \tilde{x}_0 = x_0$, constants $\tau_z = \frac{\tau_x}{\mu} - \frac{\alpha(1-\tau_x)}{\mu(L-\mu)}, \widetilde{\omega} = \sum_{k=0}^{m-1} \left(1 + \frac{\mu}{\alpha}\right)^{2k}$.

1: **for** $s = 0, \dots, S - 1$ **do**
2:      Compute and store $\nabla f(\tilde{x}_s)$.
3:      **for** $k = 0, \dots, m - 1$ **do**
4:         $y_k^s = \tau_x z_k^s + (1 - \tau_x) \tilde{x}_s + \tau_z \left(\mu(\tilde{x}_s - z_k^s) - \nabla f(\tilde{x}_s)\right)$.
5:         $z_{k+1}^s = \arg\min_x \left\{ \langle \mathcal{G}_{y_k^s}^{\text{SVRG}}, x \rangle + (\alpha/2)\|x - z_k^s\|^2 + (\mu/2)\|x - y_k^s\|^2 \right\}$.
6:      **end for**
7:      $\tilde{x}_{s+1}$ is sampled from $\left\{ P(\tilde{x}_{s+1} = y_k^s) = \frac{1}{\widetilde{\omega}} \left(1 + \frac{\mu}{\alpha}\right)^{2k} \,\middle|\, k \in \{0, \dots, m-1\} \right\}$.
8:      $z_0^{s+1} = z_m^s$.
9: **end for**

**Output:** $z_0^S$.

---

## 4.1 BS-SVRG

As mentioned in Section 2, the shifted SVRG estimator $\mathcal{H}_x^{\text{SVRG}}$ induces a practical $\mathcal{G}_x^{\text{SVRG}}$ (which is just the original SVRG estimator [20]) and thus by using Lemma 1, we obtain a practical updating rule and a classic equality for the shifted estimator. Now we can design an accelerated SVRG variant that minimizes $h$. To make the notations specific, we define $\mathcal{G}_{x_k}^{\text{SVRG}} \triangleq \nabla f_{i_k}(x_k) - \nabla f_{i_k}(\tilde{x}_s) + \nabla f(\tilde{x}_s)$, where $i_k$ is sampled uniformly in $[n]$ and $\tilde{x}_s$ is a previously chosen random anchor point. For simplicity, in what follows, we only consider constant parameter choices. We name our SVRG variant as BS-SVRG (Algorithm 2), which is designed based on the following thought experiment.

**Thought experiment.** We design BS-SVRG by extending G-TM, which is natural since almost all the existing stochastic accelerated methods are constructed based on NAG. For SVRG, its (directly) accelerated variants [1, 60, 26] all incorporate the idea of "negative" momentum, which is basically Nesterov's momentum provided by the anchor point $\tilde{x}_s$ instead of the previous iterate. Inspired by their success, we design the "momentum step" of BS-SVRG (Step 4) by replacing all the previous iterate $y_{k-1}$ in $y_k = \tau_x z_k + (1 - \tau_x)y_{k-1} + \tau_z\left(\mu(y_{k-1} - z_k) - \nabla f(y_{k-1})\right)$ with the anchor point $\tilde{x}_s$. The insight is that the "momentum step" is aggressive and could be erroneous in the stochastic case. Thus, we construct it based on some "stable" point instead of the previous stochastic iterate.

We adopt a similar Lyapunov function as G-TM: $T_s \triangleq h(\tilde{x}_s) - c_1\|\nabla h(\tilde{x}_s)\|^2 + \frac{\lambda}{2}\|z_0^s - x^\star\|^2$, where $c_1 \in \left[0, \frac{1}{2(L-\mu)}\right]$ and $\lambda > 0$ and build the per-epoch contraction of BS-SVRG as follows.

**Theorem 2.** *In Algorithm 2, if we choose $\alpha, \tau_x$ under the constraints $(1 + \frac{\mu}{\alpha})^{2m}(1 - \tau_x) \leq 1$ and $(1 + \tau_x)^2(1 - \tau_x) \geq 4\left((\frac{\alpha}{\mu} + 1) - (\frac{\alpha}{\mu} + \kappa)\tau_x\right)^2$, the per-epoch contraction $\mathbb{E}\left[T_{s+1}\right] \leq (1 + \frac{\mu}{\alpha})^{-2m}T_s$ holds with $\lambda = \frac{\alpha^2(1-\tau_x)}{\widetilde{\omega}(L-\mu)}(1 + \frac{\mu}{\alpha})^{2m}$. The expectation is taken wrt the information up to epoch s.*

In what follows, we provide a simple analytic choice that satisfies the constraints. We consider the ill-conditioned case where $\frac{m}{\kappa} \leq \frac{3}{4}$, and we fix $m = 2n$ to make it specific.[9] In this case, Allen-Zhu [1] derived an $O(\sqrt{6n\kappa} \log \frac{1}{\epsilon})$ expected iteration complexity[10] for Katyusha (cf. Theorem 2.1, [1]).

**Proposition 2.1** (Ill condition). *If $\frac{m}{\kappa} \leq \frac{3}{4}$, the choice $\alpha = \sqrt{cm\mu L} - \mu, \tau_x = (1 - \frac{1}{c\kappa})\frac{\sqrt{cm\kappa}}{\sqrt{cm\kappa}+\kappa-1}$, where $c = 2 + \sqrt{3}$, satisfies the constraints in Theorem 2.*

Using this parameter choice in Theorem 2, we obtain an $O(\sqrt{1.87n\kappa} \log \frac{1}{\epsilon})$ expected iteration complexity for BS-SVRG, which is around $1.8$ times lower than that of Katyusha.

**Remark 2.1.** *We are not aware of other parameter choices of Katyusha that have faster rates. Hu et al. [19] made an attempt based on dissipativity theory, but no explicit rate is given. To derive a*

*better choice for Katyusha, significant modification to its proof is required (for its parameter $\tau_2$), which results in complicated constraints and is thus out of the scope of this paper. We believe that there could be some computer-aided ways to find better choices for both Katyusha and BS-SVRG, which we leave for future work.*

For the other case where $\frac{m}{\kappa} > \frac{3}{4}$ (i.e., $\kappa = O(n)$), almost all the accelerated and non-accelerated incremental gradient methods perform the same, at an $O(n \log \frac{1}{\epsilon})$ oracle complexity (and is indeed fast). Hannah et al. [17] shows that by optimizing the parameters of SVRG and SARAH, a lower $O(n + \frac{n}{1+\max\{\log(n/\kappa),0\}} \log \frac{1}{\epsilon})$ oracle complexity is achievable. Due to these facts, we do not optimize the parameters for this case and provide the following proposition as a basic guarantee.

**Proposition 2.2** (Well condition). *If $\frac{m}{\kappa} > \frac{3}{4}$, by choosing $\alpha = \frac{3L}{2} - \mu, \tau_x = (1 - \frac{1}{6m})\frac{3\kappa}{5\kappa-2}$, the epochs of BS-SVRG satisfy $T_{s+1} \leq \frac{1}{2} \cdot T_s$ with $\lambda = \frac{2\alpha^2(1-\tau_x)}{\widetilde{\omega}(L-\mu)}$, which implies an $O(n \log \frac{1}{\epsilon})$ expected iteration complexity.*

There exists a special choice in the constraints: by choosing $\tau_x = \frac{\alpha+\mu}{\alpha+L}$, the second constraint always holds and this leads to $c_1 = 0$ in $T_s$. In this case, $\alpha$ can be found using numerical tools, which is summarized as follows.

**Proposition 2.3** (Numerical choice). *By fixing $\tau_x = \frac{\alpha+\mu}{\alpha+L}$, the optimal choice of $\alpha$ can be found by solving the equation $\left(1 + \frac{\mu}{\alpha}\right)^{2m}\left(1 - \frac{\alpha+\mu}{\alpha+L}\right) = 1$ using numerical tools, and this equation has a unique positive root.*

Compared with Katyusha, BS-SVRG has a simpler scheme, which only requires storing one variable vector $\{z_k\}$ and tuning 2 parameters similar to MiG [60]. Moreover, BS-SVRG achieves the fastest rate among the accelerated SVRG variants.

## 4.2 Accelerated SAGA Variant

As given in Section 2, the shifted SAGA estimator $\mathcal{H}_x^{\text{SAGA}}$ also induces a practical gradient estimator, and thus we can design an accelerated SAGA variant in a similar way. Inspired by the existing (directly) accelerated SAGA variant [58], we can design the recursion (updating rule of the table) as $\phi_{i_k}^{k+1} = \tau_x z_k + (1 - \tau_x)\phi_{i_k}^k + \tau_z\left(\mu(\frac{1}{n}\sum_{i=1}^n \phi_i^k - z_k) - \frac{1}{n}\sum_{i=1}^n \nabla f_i(\phi_i^k)\right)$. We found that for the resulting scheme, we can adopt the following Lyapunov function ($c_1 \in \left[0, \frac{1}{2(L-\mu)}\right], \lambda > 0$): $T_k = \frac{1}{n}\sum_{i=1}^n h_i(\phi_i^k) - c_1\|\frac{1}{n}\sum_{i=1}^n \nabla h_i(\phi_i^k)\|^2 + \frac{\lambda}{2}\|z_k - x^\star\|^2$, which is an "incremental version" of $T_s$. Note that $\frac{1}{n}\sum_{i=1}^n h_i(\phi_i^k) - c_1\|\frac{1}{n}\sum_{i=1}^n \nabla h_i(\phi_i^k)\|^2 \geq \frac{1}{n}\sum_{i=1}^n \left(h_i(\phi_i^k) - c_1\|\nabla h_i(\phi_i^k)\|^2\right) \geq 0$.

A similar accelerated rate can be derived for the SAGA variant and its parameter choice shows some interesting correspondence between the variants of SVRG and SAGA. Moreover, the resulting scheme does not need the tricky "doubling sampling" in [58] and thus it has a lower iteration complexity. However, since its updating rules require the knowledge of point table, the scheme has an undesirable $O(nd)$ memory complexity. We provide this variant in Appendix C.4 for interested readers.

## 5 Finite-Sum Objectives with Incremental Proximal Point Oracle

We consider the finite-sum objective (1) and assume that the proximal operator oracle $\text{prox}_i^\alpha(\cdot)$ of each $f_i$ is available. Point-SAGA [11] is a typical method that utilizes this oracle, and it achieves the same $O\left((n + \sqrt{n\kappa}) \log \frac{1}{\epsilon}\right)$ expected iteration complexity. Although in general, the incremental proximal operator oracle is much more expensive than the incremental gradient oracle, Point-SAGA is interesting in the following aspects: (1) it has a simple scheme with only 1 parameter; (2) its analysis is elegant and tight, which does not require any Young's inequality; (3) for problems where the proximal point oracle has an analytic solution, it has a very fast rate (i.e., its rate factor is smaller than $1 - (n + \sqrt{n\kappa} + 1)^{-1}$, which is faster than both Katyusha and BS-SVRG).

It might be surprising that by shifting objective, the convergence rate of Point-SAGA can be further boosted. We name the proposed variant as BS-Point-SAGA, which is presented in Algorithm 3. Recall that the Lyapunov function used to analyze Point-SAGA has the form (cf. Theorem 5, [11]) $T_k^{\text{Point-SAGA}} = \frac{c}{n}\sum_{i=1}^n \|\nabla f_i(\phi_i^k) - \nabla f_i(x^\star)\|^2 + \|x_k - x^\star\|^2$. We adopt a shifted version of this Lyapunov function (with $\lambda > 0$): $T_k = \lambda \cdot \frac{1}{n}\sum_{i=1}^n \|\nabla h_i(\phi_i^k)\|^2 + \|x_k - x^\star\|^2$. The analysis of

---

**Algorithm 3** Point-SAGA Boosted by Shifting objective (BS-Point-SAGA)

---

**Input:** Parameters $\alpha > 0$ and initial guess $x_0 \in \mathbb{R}^d$, iteration number $K$.
**Initialize:** A point table $\phi^0 \in \mathbb{R}^{d \times n}$ with $\forall i \in [n], \phi_i^0 = x_0$, running averages for the point table and its gradients.
 1: **for** $k = 0, \dots, K-1$ **do**
 2:     Sample $i_k$ uniformly in $[n]$.
 3:     Update $x$: $z_k = x_k + \frac{1}{\alpha}\left(\nabla f_{i_k}(\phi_{i_k}^k) - \frac{1}{n}\sum_{i=1}^n \nabla f_i(\phi_i^k) + \mu\left(\frac{1}{n}\sum_{i=1}^n \phi_i^k - \phi_{i_k}^k\right)\right)$,
           $x_{k+1} = \text{prox}_{i_k}^\alpha(z_k)$.
 4:     Set $\phi_{i_k}^{k+1} = x_{k+1}$ and keep other entries unchanged (i.e., for $i \neq i_k, \phi_i^{k+1} = \phi_i^k$). Update the running averages according to the change in $\phi^{k+1}$ (note that $\nabla f_{i_k}(\phi_{i_k}^{k+1}) = \alpha(z_k - x_{k+1})$).
 5: **end for**
**Output:** $x_K$.

---

BS-Point-SAGA is a direct application of Lemma 2. We build the per-iteration contraction in the following theorem, and its proof is given in Appendix D.

**Theorem 3.** *In Algorithm 3, if we choose $\alpha$ as the unique positive root of the cubic equation $2(\frac{\alpha}{\mu})^3 - (4n-6)(\frac{\alpha}{\mu})^2 - (2n\kappa + 4n - 6)(\frac{\alpha}{\mu}) - (n\kappa + n - 2) = 0$, the per-iteration contraction $\mathbb{E}_{i_k}[T_{k+1}] \leq (1 + \frac{\mu}{\alpha})^{-2} T_k$ holds with $\lambda = \frac{n}{\alpha^2} + \frac{2(\alpha+\mu)(n-1)}{\alpha^2(L-\mu)}$. The root of this cubic equation satisfies $\frac{\alpha}{\mu} = O(n + \sqrt{n\kappa})$, which implies an $O\left((n + \sqrt{n\kappa})\log\frac{1}{\epsilon}\right)$ expected iteration complexity.*

The expected worst-case rate factor of BS-Point-SAGA is minimized by solving the cubic equation in Theorem 3 exactly. The analytic solution of this equation is messy, but it can be easily calculated using numerical tools. In Figure 1, we numerically compare the rate factors of Point-SAGA and BS-Point-SAGA. When $\kappa$ is large, the rate factor of BS-Point-SAGA is close to the square of the rate factor of Point-SAGA, which implies an almost 2 times lower expected iteration complexity. In terms of memory requirement, BS-Point-SAGA has an undesirable $O(nd)$ complexity since the update of $x_{k+1}$ involves $\phi_{i_k}^k$. Nevertheless, it achieves the fastest known rate for finite-sum problems (if both $L$ and $\mu$ are known).

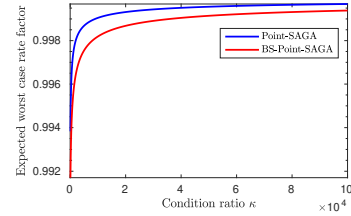

Figure 1: A comparison of the expected worst-case rate factors.

## 6 Performance Evaluations

In general, a faster worst-case rate does not necessarily imply a better empirical performance. It is possible that the slower rate is loose or the worst-case analysis is not representative of reality (e.g., worst-case scenarios are not stable to perturbations). We provide experimental results of the proposed methods in this section. We evaluate them in the ill-conditioned case where the problem has a huge $\kappa$ to justify the accelerated $\sqrt{\kappa}$ dependence. Detailed experimental setup can be found in Appendix E.

We started with evaluating the deterministic methods: NAG, TM and G-TM. We first did a simulation on the quadratic objective mentioned in Section 3.1.1, which also serves as a justification of Proposition 1.1. In this simulation, the default (constant) parameter choices were used and all the methods were initialized in $(-100, 100)$. We plot their convergences and theoretical guarantees (marked with "UB") in Figure 2a (the bound for TM is not shown due to the initial state issue). This simulation shows that after the first iteration, TM and G-TM have the same rate, and the initial state issue of TM can make it slower than NAG. It also suggests that the guarantee of NAG is loose.

Then, we measured their performance on real world datasets from LIBSVM [9]. The task we chose is $\ell_2$-logistic regression. We normalized the datasets and thus for this problem, $L = 0.25 + \mu$. For real world tasks, we tracked function value suboptimality, which is easier to compute than $\|x - x^\star\|^2$ in practice. The result is given in Figure 2b. In the first 30 iterations, TM is slower than G-TM due to the initial state issue. After that, they are almost identical and are faster than NAG.

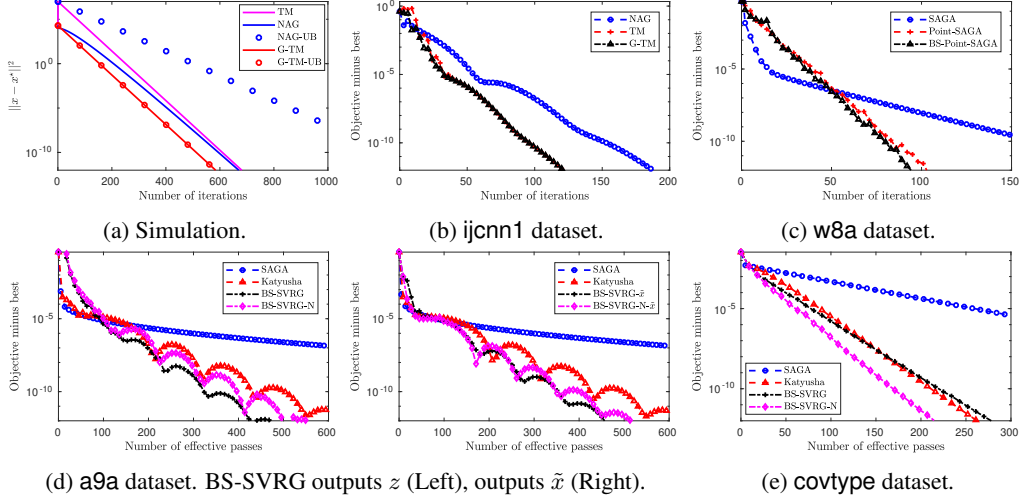

(a) Simulation.　　　　　　(b) ijcnn1 dataset.　　　　　　(c) w8a dataset.

(d) a9a dataset. BS-SVRG outputs $z$ (Left), outputs $\tilde{x}$ (Right).　　　　　(e) covtype dataset.

Figure 2: Evaluations. (a) Quadratic, $L = 1, \mu = 10^{-3}$. (b) $\ell_2$-logistic regression, $\mu = 10^{-3}$. (c) Ridge regression, $\mu = 5 \times 10^{-7}$. (d) (e) $\ell_2$-logistic regression, $\mu = 10^{-8}$.

We then evaluated BS-SVRG on the same problem, which can fully utilize the finite-sum structure. We evaluated two parameter choices of BS-SVRG: (1) the analytic choice in Proposition 2.1 (marked as "BS-SVRG"); (2) the numerical choice in Proposition 2.3 (marked as "BS-SVRG-N"). We selected SAGA ($\gamma = \frac{1}{2(\mu n + L)}$, [12]) and Katyusha ($\tau_2 = \frac{1}{2}, \tau_1 = \sqrt{\frac{m}{3\kappa}}, \alpha = \frac{1}{3\tau_1 L}$, [1]) with their default parameter choices as the baselines. Since SAGA and SVRG-like algorithms have different iteration complexities, we plot the curve with respect to the number of data passes. The results are given in Figure 2d and 2e. In the experiment on a9a dataset (Figure 2d (Left)), both choices of BS-SVRG perform well after 100 passes. The issue of their early stage performance can be eased by outputting the anchor point $\tilde{x}$ instead, as shown in Figure 2d (Right).

We also conducted an empirical comparison between BS-Point-SAGA and Point-SAGA in Figure 2c. Their analytic parameter choices were used. We chose ridge regression as the task since its proximal operator has a closed form solution (see Appendix A in [11]). For this objective, after normalizing the dataset, $L = 1 + \mu$. The performance of SAGA is also plotted as a reference.

## 7   Conclusion

In this work, we focused on unconstrained smooth strongly convex problems and designed new schemes for a shifted objective. Lemma 1 and Lemma 2 are the cornerstones for the new designs, which serve as instantiations of the shifted gradient oracle. Following this methodology, we proposed G-TM, BS-SVRG (and BS-SAGA) and BS-Point-SAGA. The new schemes achieve faster worst-case rates and have tighter and simpler proofs compared with their existing counterparts. Experiments on machine learning tasks show some improvement of the proposed methods.

Although provided only for strongly convex problems, our framework of exploiting the interpolation condition (i.e., Algorithm 1) can also be extended to the non-strongly convex case ($\mu = 0$). It can be easily verified that Theorem 1 holds with $\mu = 0$ and thus we can choose a variable-parameter setting that leads to the $O(1/K^2)$ rate. It turns out that Algorithm 1 in this case is equivalent to the optimized gradient method [21], which is also covered by the second accelerated method (14) studied in [48]. Moreover, the Lyapunov function $T_k$ becomes $a_k \left( f(y_{k-1}) - f(x^\star) - \frac{1}{2L} \|\nabla f(y_{k-1})\|^2 \right) + \frac{L}{4} \|z_k - x^\star\|^2$ for some $a_k > 0$, which is exactly the one used in Theorem 11, [48].

While the proposed approach boosts the convergence rate, some limitations should be stressed. First, it requires a prior knowledge of the strong convexity constant $\mu$ since even if it is applied to a non-accelerated method, the parameter choice is always related to $\mu$. Furthermore, this methodology relies heavily on the interpolation condition, which requires $f$ to be defined everywhere on $\mathbb{R}^d$ [13]. This restriction makes it hardly generalizable to the constrained/proximal setting [34] (for the proximal case, a possible solution is to assume that the smooth part is defined everywhere on $\mathbb{R}^d$ [7, 22, 50]).

## Broader Impact

This work studies the performance limit of solving a class of convex problem. Data scientists and machine learning researchers may benefit from this work by using the proposed methods to boost the training speed of their models. We are not aware of clear negative outcomes of this work since we focus more on the fundamental understanding of convex optimization.

## Acknowledgments and Disclosure of Funding

We thank the reviewers for their valuable and constructive comments. This work was partially supported by GRF 14208318 from the RGC and ITF 6904945 from the ITC of HKSAR.

## Footnotes

[1]The formal definitions of smoothness, strong convexity are given in Section 1.1. If each $f_i(\cdot)$ is $L$-smooth, the averaged function $f(\cdot)$ is itself $L$-smooth — but typically with a smaller $L$. We keep $L$ as the smoothness constant for consistency.

[2]It implies that those inequalities may allow a non-smooth $f$ interpolating the set, and thus a worst-case rate built upon those inequalities may not be achieved by any smooth $f$ (i.e., the rate is loose). See [51] for details.

[3]In this paper, the worst-case convergence rate is measured in terms of the squared norm distance $\|x - x^\star\|^2$.

[4]In the Lyapunov analysis framework, this is equivalent to picking a family of Lyapunov function that only involves the shifted objective $h$ (instead of $f$). See [5] for a nice review of Lyapunov-function-based proofs.

[5]In the Euclidean case, mirror descent coincides with GD. It represents another approach to the same method.

[6]In the strongly convex setting, $(1 + \frac{\mu}{\alpha}) \cdot \mathrm{prox}_i^\alpha$ is firmly non-expansive (e.g., Proposition 1 in [11]).

[7]In our opinion, the most important techniques in NAG are Lemma 3 for $f$ and the mirror descent lemma. Algorithm 1 was derived by having a shifted version of Lemma 3 for $h$ and the shifted mirror descent lemma.

[8]This is also the example where GD with $^2/_{(L+\mu)}$ stepsize behaves exactly like its worst-case analysis.

[9] We choose the setting that is used in the analysis and experiments of Katyusha [1] to make a fair comparison.

[10] We are referring to the expected number of stochastic iterations (e.g., in total $Sm$ in Algorithm 2) required to achieve $\|x - x^\star\|^2 \leq \epsilon$. If $m = 2n$, in average, each stochastic iteration of SVRG requires $1.5$ oracle calls.

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
