[Supplementary Material 1 · supp.pdf]

# Supplementary Materials for
# "Boosting First-Order Methods by Shifting Objective: New Schemes with Faster Worst-Case Rates"

## A Technical lemmas with proofs

**Lemma 1** (Shifted mirror descent lemma). *Given a gradient estimator $\mathcal{G}_y$, vectors $z^+, z^-, y \in \mathbb{R}^d$, fix the updating rule $z^+ = \arg\min_x \left\{ \langle \mathcal{G}_y, x \rangle + \alpha/2 \|x - z^-\|^2 + \mu/2 \|x - y\|^2 \right\}$. Suppose that we have a shifted gradient estimator $\mathcal{H}_y$ satisfying the relation $\mathcal{H}_y = \mathcal{G}_y - \mu(y - x^\star)$, it holds that*

$$\langle \mathcal{H}_y, z^- - x^\star \rangle = \frac{\alpha}{2}\left( \|z^- - x^\star\|^2 - \left(1 + \frac{\mu}{\alpha}\right)^2 \|z^+ - x^\star\|^2 \right) + \frac{1}{2\alpha}\|\mathcal{H}_y\|^2.$$

*Proof.* Using the optimality condition,

$$\mathcal{G}_y + \alpha(z^+ - z^-) + \mu(z^+ - y) = \mathbf{0},$$
$$\mathcal{H}_y + \alpha(z^+ - z^-) + \mu(z^+ - x^\star) = \mathbf{0},$$
$$(\alpha + \mu)(z^+ - x^\star) = \alpha(z^- - x^\star) - \mathcal{H}_y,$$
$$(\alpha + \mu)^2 \|z^+ - x^\star\|^2 = \alpha^2 \|z^- - x^\star\|^2 - 2\alpha \langle \mathcal{H}_y, z^- - x^\star \rangle + \|\mathcal{H}_y\|^2.$$

Re-arranging the last equality completes the proof. $\qquad\square$

**Lemma 2** (Shifted firm non-expansiveness). *Given relations $z^+ = \mathrm{prox}_i^\alpha(z^-)$ and $y^+ = \mathrm{prox}_i^\alpha(y^-)$, it holds that*

$$\frac{1}{\alpha^2}\left(1 + \frac{2(\alpha + \mu)}{L - \mu}\right)\|\nabla h_i(z^+) - \nabla h_i(y^+)\|^2 + \left(1 + \frac{\mu}{\alpha}\right)^2 \|z^+ - y^+\|^2 \leq \|z^- - y^-\|^2.$$

*Proof.* Based on the first-order optimality condition and the definition of $h_i$,

$$\nabla f_i(z^+) + \alpha(z^+ - z^-) = \mathbf{0}, \qquad \nabla f_i(y^+) + \alpha(y^+ - y^-) = \mathbf{0},$$
$$\nabla h_i(z^+) + \nabla f_i(x^\star) + \mu(z^+ - x^\star) + \alpha(z^+ - z^-) = \mathbf{0},$$
$$\nabla h_i(y^+) + \nabla f_i(x^\star) + \mu(y^+ - x^\star) + \alpha(y^+ - y^-) = \mathbf{0}.$$

Subtract the last two equalities,

$$(\alpha + \mu)(z^+ - y^+) = \alpha(z^- - y^-) - \left(\nabla h_i(z^+) - \nabla h_i(y^+)\right), \tag{4}$$

which implies

$$(\alpha + \mu)^2 \|z^+ - y^+\|^2 = \alpha^2 \|z^- - y^-\|^2 - 2\alpha \langle \nabla h_i(z^+) - \nabla h_i(y^+), z^- - y^- \rangle \\ + \|\nabla h_i(z^+) - \nabla h_i(y^+)\|^2. \tag{5}$$

Based on the interpolation condition of $h_i$, we have

$$\langle \nabla h_i(z^+) - \nabla h_i(y^+), z^+ - y^+ \rangle \geq \frac{1}{L - \mu}\|\nabla h_i(z^+) - \nabla h_i(y^+)\|^2.$$

Together with (4), it holds that

$$\langle \nabla h_i(z^+) - \nabla h_i(y^+), z^- - y^- \rangle \geq \frac{1}{\alpha}\left(1 + \frac{\alpha + \mu}{L - \mu}\right)\|\nabla h_i(z^+) - \nabla h_i(y^+)\|^2.$$

It remains to use this bound in (5). $\qquad\square$

Forming convex combination between vector sequences is a common technique in designing accelerated methods (e.g., [4, 25, 16, 1]). From an analytical perspective, convex combination facilitates building a contraction between function values and the coefficient directly controls the contraction ratio, which is summarized in the following lemma. Unlike previous works, we allow a residual term $\mathcal{R}$ in the convex combination.

**Lemma 3** (Function-value contraction)**.** *Given a continuously differentiable and convex function $f$, vectors $x^+, x^-, z, \mathcal{R} \in \mathbb{R}^d$ and scalar $\tau \in ]0, 1[$, if $x^+ = \tau z + (1 - \tau)x^- + \mathcal{R}$, it satisfies that*

$$f(x^+) - f(x^\star) \leq (1 - \tau)\big(f(x^-) - f(x^\star)\big) + \langle \nabla f(x^+), \mathcal{R} \rangle + \tau \langle \nabla f(x^+), z - x^\star \rangle.$$

*Proof.* Using convexity twice,

$$
\begin{aligned}
f(x^+) - f(x^\star) &\leq \langle \nabla f(x^+), x^+ - x^\star \rangle \\
&= \langle \nabla f(x^+), x^+ - z \rangle + \langle \nabla f(x^+), z - x^\star \rangle \\
&= \frac{1 - \tau}{\tau} \langle \nabla f(x^+), x^- - x^+ \rangle + \frac{1}{\tau} \langle \nabla f(x^+), \mathcal{R} \rangle + \langle \nabla f(x^+), z - x^\star \rangle \\
&\leq \frac{1 - \tau}{\tau} \big(f(x^-) - f(x^+)\big) + \frac{1}{\tau} \langle \nabla f(x^+), \mathcal{R} \rangle + \langle \nabla f(x^+), z - x^\star \rangle.
\end{aligned}
$$

Re-arranging this inequality completes the proof. $\qquad\square$

This simple trick (with $\mathcal{R} = \mathbf{0}$) appears frequently in the proofs of existing accelerated first-order methods. Note that the convexity arguments in this lemma can be strengthened by the interpolation condition or strong convexity if $f$ satisfies additional assumptions.

# B  Proofs for Section 3

## B.1  Generality of the framework of Algorithm 1

First, we show that TM is a parameterization of NAG (Algorithm 5 in Appendix F). Note that TM has the following scheme (the notations follow the ones in [10]):

$$
\begin{aligned}
x_{k+1} &= x_k + \beta(x_k - x_{k-1}) - \alpha \nabla f(y_k), \\
y_{k+1} &= x_{k+1} + \gamma(x_{k+1} - x_k), \\
z_{k+1} &= x_{k+1} + \delta(x_{k+1} - x_k).
\end{aligned}
$$

By casting this scheme into the framework of Algorithm 5, we obtain

$$
\begin{aligned}
y_k &= \frac{\gamma}{\delta} z_k + \left(1 - \frac{\gamma}{\delta}\right) x_k, \\
z_{k+1} &= \frac{\beta(1 + \delta) - \gamma}{\delta - \gamma} z_k + \frac{\delta - \beta(1 + \delta)}{\delta - \gamma} y_k - \alpha(1 + \delta)\nabla f(y_k), \\
x_{k+1} &= \frac{1}{1 + \delta} z_{k+1} + \frac{\delta}{1 + \delta} x_k.
\end{aligned}
$$

Substituting the parameter choice of TM, we see that TM is equivalent to choosing $\alpha = \sqrt{L\mu} - \mu, \tau_y = (\sqrt{\kappa} + 1)^{-1}, \tau_x = \frac{2\sqrt{\kappa} - 1}{\kappa}$ in Algorithm 5. Interestingly, this choice and the choice of NAG (given in Appendix F) only differ in $\tau_x$.

Then, we show that Algorithm 5 is an instance of the framework of Algorithm 1. By expanding the convex combinations of sequences $\{y_k\}$ and $\{x_k\}$ in Algorithm 5, we can conclude that

$$y_k = \tau_x z_k + (1 - \tau_x)y_{k-1} + \tau_y(1 - \tau_x)(z_k - z_{k-1}).$$

Based on the optimality condition at iteration $k - 1$, we have

$$\alpha(z_k - z_{k-1}) = \mu(y_{k-1} - z_k) - \nabla f(y_{k-1}).$$

Now, it is clear that Algorithm 5 is an instance of the framework of Algorithm 1 with the variable-parameter choice (let $y_{-1} = x_0$): at $k = 0, \tau_0^x = \tau_y, \tau_0^z = 0$; at $k \geq 1, \tau_k^x = \tau_x, \tau_k^z = \frac{\tau_y(1 - \tau_x)}{\alpha}$.

## B.2 Proof of Theorem 1

First, we can introduce a contraction between $h(y_k)$ and $h(y_{k-1})$ using Lemma 3. Applying Lemma 3 with $f = h$ for the recursion $y_k = \tau_k^x z_k + (1 - \tau_k^x) y_{k-1} + \tau_k^z \big(\mu(y_{k-1} - z_k) - \nabla f(y_{k-1})\big)$ and strengthening the convexity arguments by the interpolation condition, we obtain

$$h(y_k) \leq (1 - \tau_k^x) h(y_{k-1}) + \tau_k^z \langle \nabla h(y_k), \mu(y_{k-1} - z_k) - \nabla f(y_{k-1}) \rangle + \tau_k^x \langle \nabla h(y_k), z_k - x^\star \rangle$$
$$- \frac{\tau_k^x}{2(L - \mu)} \|\nabla h(y_k)\|^2 - \frac{1 - \tau_k^x}{2(L - \mu)} \|\nabla h(y_{k-1}) - \nabla h(y_k)\|^2.$$

Note that $\mu(y_{k-1} - z_k) - \nabla f(y_{k-1}) = \mu(x^\star - z_k) - \nabla h(y_{k-1})$ by definition, and thus

$$h(y_k) \leq (1 - \tau_k^x) h(y_{k-1}) - \tau_k^z \langle \nabla h(y_k), \nabla h(y_{k-1}) \rangle + (\tau_k^x - \mu\tau_k^z) \langle \nabla h(y_k), z_k - x^\star \rangle$$
$$- \frac{\tau_k^x}{2(L - \mu)} \|\nabla h(y_k)\|^2 - \frac{1 - \tau_k^x}{2(L - \mu)} \|\nabla h(y_{k-1}) - \nabla h(y_k)\|^2. \tag{6}$$

Then, to build a contraction between $\|z_{k+1} - x^\star\|^2$ and $\|z_k - x^\star\|^2$, we apply Lemma 1 with $\mathcal{G}_y = \nabla f(y_k), \mathcal{H}_y = \nabla h(y_k)$ and $z^+ = z_{k+1}$, which gives

$$\langle \nabla h(y_k), z_k - x^\star \rangle = \frac{\alpha_k}{2} \left( \|z_k - x^\star\|^2 - \left(1 + \frac{\mu}{\alpha_k}\right)^2 \|z_{k+1} - x^\star\|^2 \right) + \frac{1}{2\alpha_k} \|\nabla h(y_k)\|^2.$$

Using this relation in (6), expanding and re-arranging the terms, we conclude that

$$h(y_k) - \left( \frac{\tau_k^x - \mu\tau_k^z}{2\alpha_k} - \frac{1}{2(L - \mu)} \right) \|\nabla h(y_k)\|^2 + \frac{\alpha_k(\tau_k^x - \mu\tau_k^z)}{2} \left(1 + \frac{\mu}{\alpha_k}\right)^2 \|z_{k+1} - x^\star\|^2$$

$$\leq (1 - \tau_k^x) \left( h(y_{k-1}) - \frac{1}{2(L - \mu)} \|\nabla h(y_{k-1})\|^2 \right) + \frac{\alpha_k(\tau_k^x - \mu\tau_k^z)}{2} \|z_k - x^\star\|^2$$

$$+ \left( \frac{1 - \tau_k^x}{L - \mu} - \tau_k^z \right) \langle \nabla h(y_k), \nabla h(y_{k-1}) \rangle.$$

It remains to impose parameter constraints according to the Lyapunov function.

## B.3 Proof of Proposition 1.1

First, we can write the $k$th-update of G-TM with constant parameter as

$$y_k = (\tau_x - \tau_z\mu)z_k + \big(1 - (\tau_x - \tau_z\mu)\big)y_{k-1} - \tau_z\nabla f(y_{k-1}),$$

$$z_{k+1} = \frac{\alpha}{\alpha + \mu}z_k + \frac{\mu}{\alpha + \mu}y_k - \frac{1}{\alpha + \mu}\nabla f(y_k).$$

Substituting the constant parameter choice, we obtain

$$y_k = \frac{2}{\sqrt{\kappa} + 1}z_k + \frac{\sqrt{\kappa} - 1}{\sqrt{\kappa} + 1}\left( y_{k-1} - \frac{1}{L}\nabla f(y_{k-1}) \right),$$

$$z_{k+1} = \left(1 - \frac{1}{\sqrt{\kappa}}\right)z_k + \frac{1}{\sqrt{\kappa}}y_k - \frac{1}{\sqrt{L\mu}}\nabla f(y_k).$$

For the objective function $f(x) = \frac{1}{2} \left\langle \begin{bmatrix} L & 0 \\ 0 & \mu \end{bmatrix} x, x \right\rangle$, the update can be further expanded as

$$y_k = \frac{2}{\sqrt{\kappa} + 1}z_k + \begin{bmatrix} 0 & 0 \\ 0 & \frac{(\sqrt{\kappa} - 1)^2}{\kappa} \end{bmatrix} y_{k-1},$$

$$z_{k+1} = \left(1 - \frac{1}{\sqrt{\kappa}}\right)z_k + \begin{bmatrix} -\frac{\kappa - 1}{\sqrt{\kappa}} & 0 \\ 0 & 0 \end{bmatrix} y_k.$$

Thus,

$$z_{k+1} = \left(1 - \frac{1}{\sqrt{\kappa}}\right) \begin{bmatrix} -1 & 0 \\ 0 & 1 \end{bmatrix} z_k \implies \|z_{k+1} - x^\star\|^2 = \left(1 - \frac{1}{\sqrt{\kappa}}\right)^2 \|z_k - x^\star\|^2,$$

as desired.

## C   Proofs for Section 4

### C.1   Proof of Theorem 2

For simplicity of presentation, we omit the superscript $s$ for iterates in the same epoch.

Using the trick in Lemma 3 for the recursion $y_k = \tau_x z_k + (1 - \tau_x)\,\tilde{x}_s + \tau_z\,(\mu(\tilde{x}_s - z_k) - \nabla f(\tilde{x}_s))$ and strengthening the convexity arguments by interpolation condition, we obtain

$$
h(y_k) \leq \frac{1 - \tau_x}{\tau_x}\,\langle \nabla h(y_k), \tilde{x}_s - y_k \rangle + \frac{\tau_z}{\tau_x}\,\langle \nabla h(y_k), \mu(\tilde{x}_s - z_k) - \nabla f(\tilde{x}_s) \rangle + \langle \nabla h(y_k), z_k - x^\star \rangle
$$
$$
- \frac{1}{2(L - \mu)}\|\nabla h(y_k)\|^2.
$$

Note that here the inner product $\langle \nabla h(y_k), \tilde{x}_s - y_k \rangle$ is not upper bounded as before. This term is preserved to deal with the variance.

By the definition of $h$, $\mu(\tilde{x}_s - z_k) - \nabla f(\tilde{x}_s) = \mu(x^\star - z_k) - \nabla h(\tilde{x}_s)$. Applying Lemma 1 with $\mathcal{H}_y = \mathcal{H}_{y_k}^{\mathrm{SVRG}}, \mathcal{G}_y = \mathcal{G}_{y_k}^{\mathrm{SVRG}}, z^+ = z_{k+1}$ and taking the expectation, we can conclude that

$$
h(y_k) \leq \frac{1 - \tau_x}{\tau_x}\,\langle \nabla h(y_k), \tilde{x}_s - y_k \rangle - \frac{\tau_z}{\tau_x}\,\langle \nabla h(y_k), \nabla h(\tilde{x}_s) \rangle - \frac{1}{2(L - \mu)}\|\nabla h(y_k)\|^2
$$
$$
+ \left(1 - \frac{\mu \tau_z}{\tau_x}\right)\frac{\alpha}{2}\left(\|z_k - x^\star\|^2 - \left(1 + \frac{\mu}{\alpha}\right)^2 \mathbb{E}_{i_k}\left[\|z_{k+1} - x^\star\|^2\right]\right)
$$
$$
+ \left(\frac{1}{2\alpha} - \frac{\mu \tau_z}{2\alpha \tau_x}\right)\mathbb{E}_{i_k}\left[\|\mathcal{H}_{y_k}^{\mathrm{SVRG}}\|^2\right].
$$

To bound the shifted moment, we apply the interpolation condition of $h_{i_k}$, i.e.,

$$
\mathbb{E}_{i_k}\left[\|\mathcal{H}_{y_k}^{\mathrm{SVRG}}\|^2\right] = \mathbb{E}_{i_k}\left[\|\nabla h_{i_k}(y_k) - \nabla h_{i_k}(\tilde{x}_s)\|^2\right] + 2\langle \nabla h(y_k), \nabla h(\tilde{x}_s) \rangle - \|\nabla h(\tilde{x}_s)\|^2
$$
$$
\leq 2(L - \mu)\big(h(\tilde{x}_s) - h(y_k) - \langle \nabla h(y_k), \tilde{x}_s - y_k \rangle\big) + 2\langle \nabla h(y_k), \nabla h(\tilde{x}_s) \rangle
$$
$$
- \|\nabla h(\tilde{x}_s)\|^2.
$$

After re-arranging the terms, we obtain

$$
h(y_k) \leq \left(1 - \frac{\mu \tau_z}{\tau_x}\right)\frac{L - \mu}{\alpha}\big(h(\tilde{x}_s) - h(y_k)\big)
$$
$$
+ \left[\frac{1 - \tau_x}{\tau_x} - \left(1 - \frac{\mu \tau_z}{\tau_x}\right)\frac{L - \mu}{\alpha}\right]\langle \nabla h(y_k), \tilde{x}_s - y_k \rangle
$$
$$
+ \left(1 - \frac{\mu \tau_z}{\tau_x}\right)\frac{\alpha}{2}\left(\|z_k - x^\star\|^2 - \left(1 + \frac{\mu}{\alpha}\right)^2 \mathbb{E}_{i_k}\left[\|z_{k+1} - x^\star\|^2\right]\right)
$$
$$
+ \left(\frac{1}{\alpha} - \frac{\mu \tau_z}{\alpha \tau_x} - \frac{\tau_z}{\tau_x}\right)\langle \nabla h(y_k), \nabla h(\tilde{x}_s) \rangle - \frac{1}{2(L - \mu)}\|\nabla h(y_k)\|^2
$$
$$
- \left(\frac{1}{2\alpha} - \frac{\mu \tau_z}{2\alpha \tau_x}\right)\|\nabla h(\tilde{x}_s)\|^2.
$$

To cancel $\langle \nabla h(y_k), \tilde{x}_s - y_k \rangle$, we choose $\tau_z$ such that $\frac{1 - \tau_x}{\tau_x} = \left(1 - \frac{\mu \tau_z}{\tau_x}\right)\frac{L - \mu}{\alpha}$, which gives

$$
h(y_k) \leq (1 - \tau_x)h(\tilde{x}_s) + \frac{\alpha^2(1 - \tau_x)}{2(L - \mu)}\left(\|z_k - x^\star\|^2 - \left(1 + \frac{\mu}{\alpha}\right)^2 \mathbb{E}_{i_k}\left[\|z_{k+1} - x^\star\|^2\right]\right)
$$
$$
+ \frac{\alpha + \mu - (\alpha + L)\tau_x}{(L - \mu)\mu}\,\langle \nabla h(y_k), \nabla h(\tilde{x}_s) \rangle - \frac{\tau_x}{2(L - \mu)}\|\nabla h(y_k)\|^2 \qquad (7)
$$
$$
- \frac{1 - \tau_x}{2(L - \mu)}\|\nabla h(\tilde{x}_s)\|^2.
$$

In view of the Lyapunov function $T_s \triangleq h(\tilde{x}_s) - c_1\|\nabla h(\tilde{x}_s)\|^2 + \frac{\lambda}{2}\|z_0^s - x^\star\|^2$, there are two ways to deal with the inner product $\langle \nabla h(y_k), \nabla h(\tilde{x}_s) \rangle$:

**Case I ($c_1 = 0$):** Choosing $\tau_x$ such that $\alpha + \mu - (\alpha + L)\tau_x = 0 \implies \tau_x = \frac{\alpha + \mu}{\alpha + L}$ and dropping the negative gradient norms in (7), we arrive at (9) with $c_1 = 0$.

**Case II ($c_1 \neq 0$):** Denoting $\gamma = \frac{|\alpha + \mu - (\alpha + L)\tau_x|}{(L - \mu)\mu}$ and using Young's inequality for $\langle \nabla h(y_k), \nabla h(\tilde{x}_s) \rangle$ with parameter $\beta > 0$, we can bound (7) as

$$
\begin{aligned}
h(y_k) \leq (1 - \tau_x)h(\tilde{x}_s) &+ \frac{\alpha^2(1 - \tau_x)}{2(L - \mu)}\left( \|z_k - x^\star\|^2 - \left(1 + \frac{\mu}{\alpha}\right)^2 \mathbb{E}_{i_k}\left[\|z_{k+1} - x^\star\|^2\right] \right) \\
&+ \left( \frac{\beta\gamma}{2} - \frac{\tau_x}{2(L - \mu)} \right)\|\nabla h(y_k)\|^2 - \left( \frac{1 - \tau_x}{2(L - \mu)} - \frac{\gamma}{2\beta} \right)\|\nabla h(\tilde{x}_s)\|^2.
\end{aligned}
\tag{8}
$$

We require $\gamma \neq 0$ and choose $\beta > 0$ such that

$$
\frac{\beta\gamma}{2} - \frac{\tau_x}{2(L - \mu)} = \frac{1}{1 - \tau_x}\left( \frac{1 - \tau_x}{2(L - \mu)} - \frac{\gamma}{2\beta} \right) = c_1 > 0.
$$

It can be verified that this requirement and the existence of $\beta$ are equivalent to the following constraints:

$$
\begin{cases}
\tau_x \neq \frac{\alpha + \mu}{\alpha + L}, \\
(1 + \tau_x)^2(1 - \tau_x) \geq 4\left( \left(\frac{\alpha}{\mu} + 1\right) - \left(\frac{\alpha}{\mu} + \kappa\right)\tau_x \right)^2.
\end{cases}
$$

Under these constraints, denoting $\Delta = \frac{(1 + \tau_x)^2}{(L - \mu)^2} - \frac{4\gamma^2}{1 - \tau_x} \geq 0$, we can choose $\beta = \frac{1 + \tau_x}{2\gamma(L - \mu)} + \frac{\sqrt{\Delta}}{2\gamma}$, which ensures $c_1 \in \left] 0, \frac{1}{2(L - \mu)} \right[$.

Let $c_2 \triangleq \frac{\alpha^2(1 - \tau_x)}{L - \mu}$. These two cases result in the same inequality:

$$
\begin{aligned}
h(y_k) - c_1\|\nabla h(y_k)\|^2 \leq (1 - \tau_x)\big( h(\tilde{x}_s) &- c_1\|\nabla h(\tilde{x}_s)\|^2 \big) \\
&+ \frac{c_2}{2}\left( \|z_k - x^\star\|^2 - \left(1 + \frac{\mu}{\alpha}\right)^2 \mathbb{E}_{i_k}\left[\|z_{k+1} - x^\star\|^2\right] \right).
\end{aligned}
\tag{9}
$$

Finally, summing the above inequality from $k = 0, \ldots, m - 1$ with weight $\left(1 + \frac{\mu}{\alpha}\right)^{2k}$, we conclude that

$$
\begin{aligned}
\mathbb{E}\big[ h(\tilde{x}_{s+1}) - c_1\|\nabla h(\tilde{x}_{s+1})\|^2 \big] &= \sum_{k=0}^{m-1} \frac{1}{\widetilde{\omega}}\left(1 + \frac{\mu}{\alpha}\right)^{2k} \mathbb{E}\big[ h(y_k^s) - c_1\|\nabla h(y_k^s)\|^2 \big] \\
&\leq (1 - \tau_x)\big( h(\tilde{x}_s) - c_1\|\nabla h(\tilde{x}_s)\|^2 \big) + \frac{c_2}{2\widetilde{\omega}}\left( \|z_0^s - x^\star\|^2 - \left(1 + \frac{\mu}{\alpha}\right)^{2m} \mathbb{E}\big[\|z_m^s - x^\star\|^2\big] \right).
\end{aligned}
\tag{10}
$$

Imposing the constraint $\left(1 + \frac{\mu}{\alpha}\right)^{2m}(1 - \tau_x) \leq 1$ completes the proof.

### C.2 Proof of Proposition 2.1

The choice

$$
\begin{cases}
\alpha = \sqrt{cm\mu L} - \mu, \\
\tau_x = \left(1 - \frac{1}{c\kappa}\right)\frac{\alpha + \mu}{\alpha + L} = \left(1 - \frac{1}{c\kappa}\right)\frac{\sqrt{cm\kappa}}{\sqrt{cm\kappa} + \kappa - 1},
\end{cases}
$$

and the constraints

$$
(1 + \tau_x)^2(1 - \tau_x) \geq 4\left( \left(\frac{\alpha}{\mu} + 1\right) - \left(\frac{\alpha}{\mu} + \kappa\right)\tau_x \right)^2,
\tag{11}
$$

$$
\left(1 + \frac{\mu}{\alpha}\right)^{2m}(1 - \tau_x) \leq 1,
\tag{12}
$$

are put here for reference.

Note that for $m \in \left(0, \frac{3}{4}\kappa\right]$, $\tau_x = \frac{c\kappa - 1}{c\kappa + \sqrt{\frac{c\kappa}{m}}(\kappa - 1)}$ increases monotonically and $\frac{1+\tau_x}{m}$ decreases monotonically as $m$ increases. Thus, for the constraint (11), letting

$$\phi(m, \kappa) \triangleq \frac{(1 + \tau_x)^2 (1 - \tau_x)}{\left(\left(\frac{\alpha}{\mu} + 1\right) - \left(\frac{\alpha}{\mu} + \kappa\right)\tau_x\right)^2} = \frac{1 + \tau_x}{m}\left(1 - \tau_x^2\right)c\kappa,$$

we have $\phi(m, \kappa)$ decreases monotonically as $m$ increases.

When $m = \frac{3}{4}\kappa$, $\tau_x = \frac{c\kappa - 1}{\left(c + \sqrt{\frac{4c}{3}}\right)\kappa - \sqrt{\frac{4c}{3}}}$. For $\kappa \geq 1$, if $c + \sqrt{\frac{4c}{3}} - c\sqrt{\frac{4c}{3}} \leq 0 \Leftrightarrow c \geq \frac{(\sqrt{3}+\sqrt{19})^2}{16} \approx$ 2.319, we have $\tau_x$ decreases monotonically as $\kappa$ increases. In this case, letting $\kappa \to \infty$, we conclude that $\tau_x > \frac{c}{c + \sqrt{\frac{4c}{3}}} > \frac{1}{3}$, which implies that $(1 + \tau_x)^2(1 - \tau_x)$ increases monotonically as $\tau_x$ decreases. Thus,

$$\phi(m, \kappa) \geq \phi\left(\frac{3}{4}\kappa, \kappa\right) \geq \phi\left(\frac{3}{4}, 1\right) = \frac{4}{3}\left(1 + \frac{c-1}{c}\right)\left(1 - \left(\frac{c-1}{c}\right)^2\right)c.$$

To meet the constraint (11), we require $c \geq 2 + \sqrt{3} \approx 3.74$.

For constraint (12), defining

$$\psi(m, \kappa) \triangleq \left(\frac{\alpha + \mu}{\alpha}\right)^{2m}(1 - \tau_x) = \left(1 + \frac{1}{\sqrt{cm\kappa} - 1}\right)^{2m}\frac{\sqrt{cm\kappa} + c\kappa(\kappa - 1)}{(\sqrt{cm\kappa} - 1 + \kappa)c\kappa},$$

we have $\frac{\partial \psi}{\partial m} =$

$$\left(1 + \frac{1}{\sqrt{cm\kappa} - 1}\right)^{2m}\left[\left(2\ln\left(1 + \frac{1}{\sqrt{cm\kappa} - 1}\right) - \frac{1}{\sqrt{cm\kappa} - 1}\right)\frac{\sqrt{cm\kappa} + c\kappa(\kappa - 1)}{(\sqrt{cm\kappa} - 1 + \kappa)c\kappa}\right.$$
$$\left. - \frac{(\kappa - 1)(c\kappa - 1)}{2\sqrt{cm\kappa}(\sqrt{cm\kappa} - 1 + \kappa)^2}\right].$$

Denote $q = \sqrt{cm\kappa} - 1 > 0$. The roots of $\frac{\partial \psi}{\partial m}$ are identified by the following equation:

$$s(q) \triangleq 2\ln\left(1 + \frac{1}{q}\right) - \frac{1}{q} - \frac{b_0}{(q+1)(q+\kappa)(q+b_1)} = 0,$$

where $b_0 = \frac{c\kappa}{2}(\kappa - 1)(c\kappa - 1)$, $b_1 = 1 + c\kappa(\kappa - 1)$. Taking derivative, we see that when $q \to 0$, $s'(q) \geq \frac{1}{q^2} - \frac{2}{q(1+q)} \to \infty$. We can arrange the equation $s'(q) = 0$ as finding the real roots of a polynomial. By Descartes' rule of signs, this equation has exactly one positive root (with $c \geq 2 + \sqrt{3}$, we have $\kappa b_1 - 1 - b_0 \leq 0$ for any $\kappa \geq 1$ and then there is exactly one sign change in the polynomial). Thus, as $q$ increases, $s(q)$ first increases monotonically to the unique root and then decreases monotonically.

To see that $s(q)$ has exactly one root, let $q \to 0$, $s(q) \leq 2\ln\left(1 + \frac{1}{q}\right) - \frac{1}{q} \to -\infty$; when $q$ is large enough (e.g., $q > 2$ and $(q + \kappa)(q + b_1) > 2b_0$), $s(q) > 0$; let $q \to \infty$, $s(q) \to 0$. These facts suggest that $s(q)$ has a unique root. Thus, we conclude that, as $m$ increases, $\psi(m, \kappa)$ first decreases monotonically to the unique root and then increases monotonically, which means that for $m \in [2, \frac{3}{4}\kappa]$, $\psi(m, \kappa) \leq \max\left\{\psi(2, \kappa), \psi\left(\frac{3}{4}\kappa, \kappa\right)\right\}$.

For $\psi(2, \kappa)$, $\psi'(2, \kappa) = \left(1 + \frac{1}{\sqrt{2c\kappa} - 1}\right)^4\left(\sqrt{2c\kappa} + \kappa - 1\right)^{-2}\left(\sqrt{2c\kappa} - 1\right)^{-1}\ell(\kappa)$, where $\ell(\kappa)$ is a polynomial:

$$\ell(\kappa) \triangleq (c - 2)\kappa - \frac{5\sqrt{2c}}{2}\kappa^{\frac{1}{2}} + (c + 1) - \left(\sqrt{\frac{c}{2}} + \frac{1}{\sqrt{2c}}\right)\kappa^{-\frac{1}{2}} - 3\kappa^{-1} + \frac{3}{\sqrt{2c}}\kappa^{-\frac{3}{2}}.$$

It can be verified that with $c \geq 2 + \sqrt{3}$, for any $\kappa \geq \frac{8}{3}$, $\ell(\kappa) > 0$, which suggests that $\psi(2, \kappa) \leq \max\left\{\psi\left(2, \frac{8}{3}\right), \psi(2, \infty)\right\} \leq 1$ (with $c \geq 2 + \sqrt{3}$, $\psi\left(2, \frac{8}{3}\right) \leq 0.953$ and $\psi(2, \infty) = 1$).

For $\psi\left(\frac{3}{4}\kappa,\kappa\right)$, $\psi'\left(\frac{3}{4}\kappa,\kappa\right) = \left(1+\frac{2}{\sqrt{3c\kappa}-2}\right)^{\frac{3}{2}\kappa}\left(\left(c+\sqrt{\frac{4c}{3}}\right)\kappa - \sqrt{\frac{4c}{3}}\right)^{-1}\omega_1(\kappa)$, where

$$\omega_1(\kappa) \triangleq \left(\ln\left(1+\frac{2}{\sqrt{3c\kappa}-2}\right) - \frac{2}{\sqrt{3c\kappa}-2}\right)\left(\sqrt{3c\kappa}-\sqrt{3c}+\frac{3}{2}\right) + \frac{\sqrt{\frac{4c}{3}}c - c - \sqrt{\frac{4c}{3}}}{\left(c+\sqrt{\frac{4c}{3}}\right)\kappa - \sqrt{\frac{4c}{3}}}.$$

Let $p = \sqrt{3c}\kappa - 2 > 0$, the roots of $\omega_1(\kappa)$ are determined by the equation

$$\omega_2(p) \triangleq \ln\left(1+\frac{2}{p}\right) - \frac{2}{p} + \frac{\frac{3}{2+\sqrt{3c}}\left(\sqrt{\frac{4c}{3}}c-c-\sqrt{\frac{4c}{3}}\right)}{\left(p+\frac{4}{2+\sqrt{3c}}\right)\left(p+\frac{7}{2}-\sqrt{3c}\right)} = 0.$$

To ensure that $\omega_2(p)$ increases monotonically as $p$ increases, it suffices to set $c \leq 3.817$ (which ensures that $\omega_2'(p) > 0$). Thus, for any $p > 0$, $\omega_2(p) \leq \lim_{p\to\infty}\omega_2(p) = 0 \Rightarrow$ for any $\kappa \geq 1$, $\omega_1(\kappa) \leq 0$. Finally, we conclude that with $3.817 \geq c \geq 2+\sqrt{3}$, $\psi\left(\frac{3}{4}\kappa,\kappa\right) \leq \psi\left(2,\frac{8}{3}\right) \leq 0.953$, which completes the proof.

### C.3 Proof of Proposition 2.2

The choice $\begin{cases} \alpha = \frac{3L}{2}-\mu, \\ \tau_x = \left(1-\frac{1}{6m}\right)\frac{\alpha+\mu}{\alpha+L} = \left(1-\frac{1}{6m}\right)\frac{3\kappa}{5\kappa-2}, \end{cases}$ is put here for reference.

We examine the constraint $(1+\tau_x)^2(1-\tau_x) \geq 4\left(\left(\frac{\alpha}{\mu}+1\right) - \left(\frac{\alpha}{\mu}+\kappa\right)\tau_x\right)^2$. Let

$$\phi(m,\kappa) \triangleq \frac{(1+\tau_x)^2(1-\tau_x)}{4\left(\left(\frac{\alpha}{\mu}+1\right)-\left(\frac{\alpha}{\mu}+\kappa\right)\tau_x\right)^2} = \frac{(1+\tau_x)^2(1-\tau_x)4m^2}{\kappa^2}.$$

For $m \geq \frac{3}{4}\kappa$, we have $\tau_x$ and $(1-\tau_x)m$ increases monotonically as $m$ increases. Thus, $\phi(m,\kappa)$ increases as $m$ increases $\Longrightarrow \phi(m,\kappa) \geq \phi(\frac{3}{4}\kappa,\kappa)$.

$\phi(\frac{3}{4}\kappa,\kappa) = \frac{9}{4}(1+\tau_x)^2(1-\tau_x)$ and $\tau_x = \frac{9\kappa-2}{15\kappa-6}$ in this case. Note that for $\kappa \geq 1$, $\tau_x$ decreases as $\kappa$ increases and let $\kappa \to \infty$, we conclude that $\tau_x > \frac{3}{5} > \frac{1}{3} \Longrightarrow (1+\tau_x)^2(1-\tau_x)$ increases as $\tau_x$ decreases. Thus, $\phi(\frac{3}{4}\kappa,\kappa) \geq \phi(\frac{3}{4},1) > 1$, the constraint is satisfied.

Using this choice, we can write the per-epoch contraction (10) in Theorem 2 as

$$\mathbb{E}\left[h(\tilde{x}_{s+1}) - c_1\|\nabla h(\tilde{x}_{s+1})\|^2\right] + \frac{\alpha^2(1-\tau_x)}{2\widetilde{\omega}(L-\mu)}\left(1+\frac{\mu}{\alpha}\right)^{2m}\mathbb{E}\left[\|z_0^{s+1}-x^\star\|^2\right]$$
$$\leq (1-\tau_x)\left(h(\tilde{x}_s) - c_1\|\nabla h(\tilde{x}_s)\|^2\right) + \frac{\alpha^2(1-\tau_x)}{2\widetilde{\omega}(L-\mu)}\|z_0^s-x^\star\|^2.$$

Note that for $\frac{m}{\kappa} > \frac{3}{4}$, $\tau_x > \frac{1}{2}$ and by Bernoulli's inequality, $\left(1+\frac{\mu}{\alpha}\right)^{2m} \geq 1+\frac{2m\mu}{\alpha} = 1+\frac{4m}{3\kappa-2} > 2$. Let $\lambda = \frac{2\alpha^2(1-\tau_x)}{\widetilde{\omega}(L-\mu)}$. The above contraction becomes

$$\mathbb{E}\left[h(\tilde{x}_{s+1}) - c_1\|\nabla h(\tilde{x}_{s+1})\|^2\right] + \frac{\lambda}{2}\mathbb{E}\left[\|z_0^{s+1}-x^\star\|^2\right]$$
$$\leq \frac{1}{2}\cdot\left(h(\tilde{x}_s) - c_1\|\nabla h(\tilde{x}_s)\|^2 + \frac{\lambda}{2}\|z_0^s-x^\star\|^2\right).$$

Telescoping this inequality from $S-1$ to $0$, we obtain $T_S \leq \frac{1}{2^S}T_0$, and since $m = 2n$, these imply an $O(n\log\frac{1}{\epsilon})$ iteration complexity.

---

**Algorithm 4** SAGA Boosted by Shifting objective (BS-SAGA)

---

**Input:** Parameters $\alpha > 0, \tau_x \in\ ]0,1[$ and initial guess $x_0 \in \mathbb{R}^d$, iteration number $K$.
**Initialize:** $z_0 = x_0, \tau_z = \frac{\tau_x}{\mu} - \frac{\alpha(1-\tau_x)}{\mu(L-\mu)}$, a point table $\phi^0 \in \mathbb{R}^{d \times n}$ with $\forall i \in [n], \phi_i^0 = x_0$, running averages for the point table and its gradients.
1: **for** $k = 0, \dots, K - 1$ **do**
2:     Sample $i_k$ uniformly in $[n]$, set $\phi_{i_k}^{k+1} = \tau_x z_k + (1 - \tau_x)\,\phi_{i_k}^k + \tau_z\left(\mu(\bar{\phi}^k - z_k) - \frac{1}{n}\sum_{i=1}^n \nabla f_i(\phi_i^k)\right)$ and keep other entries unchanged (i.e., for $i \neq i_k, \phi_i^{k+1} = \phi_i^k$).
3:     $z_{k+1} = \arg\min_x \left\{ \langle \mathcal{G}_{\phi_{i_k}^{k+1}}^{\text{SAGA}}, x \rangle + (\alpha/2)\|x - z_k\|^2 + (\mu/2)\|x - \phi_{i_k}^{k+1}\|^2 \right\}.$
4:     Update the running averages according to the change in $\phi^{k+1}$.
5: **end for**
**Output:** $z_K$.

---

## C.4 BS-SAGA

To make the notations specific, we define

$$\mathcal{H}_{x_k}^{\text{SAGA}} \triangleq \nabla h_{i_k}(x_k) - \nabla h_{i_k}(\phi_{i_k}^k) + \frac{1}{n}\sum_{i=1}^n \nabla h_i(\phi_i^k)$$

$$\Rightarrow \mathcal{G}_{x_k}^{\text{SAGA}} \triangleq \nabla f_{i_k}(x_k) - \nabla f_{i_k}(\phi_{i_k}^k) + \frac{1}{n}\sum_{i=1}^n \nabla f_i(\phi_i^k) - \mu\left(\bar{\phi}^k - \phi_{i_k}^k\right),$$

where $\phi^k \in \mathbb{R}^{d \times n}$ is a point table that stores $n$ previously chosen random anchor points and $\bar{\phi}^k \triangleq \frac{1}{n}\sum_{i=1}^n \phi_i^k$ denotes the average of point table.

The Lyapunov function (with $c_1 \in \left[0, \frac{1}{2(L-\mu)}\right], \lambda > 0$) is put here for reference:

$$T_k = \frac{1}{n}\sum_{i=1}^n h_i(\phi_i^k) - c_1 \left\| \frac{1}{n}\sum_{i=1}^n \nabla h_i(\phi_i^k) \right\|^2 + \frac{\lambda}{2}\|z_k - x^\star\|^2. \tag{13}$$

We present the SAGA variant in Algorithm 4. In the following theorem, we only consider a simple case with $c_1 = 0$ in $T_k$. It is possible to analyze BS-SAGA with $c_1 \neq 0$ as is the case for BS-SVRG (the analysis in Appendix C.1). However, it leads to highly complicated parameter constraints. We provide a simple parameter choice similar to the one in Proposition 2.3.

**Theorem C.1.** *In Algorithm 4, if we choose $\alpha, \tau_x$ as*

$$\begin{cases} \alpha \text{ is solved from the equation } \left(1 + \frac{\mu}{\alpha}\right)^2 \left(1 - \frac{\alpha+\mu}{(\alpha+L)n}\right) = 1, \\ \tau_x = \frac{\alpha+\mu}{\alpha+L}, \end{cases} \tag{14}$$

*the following per-iteration contraction holds for the Lyapunov function defined at (13) (with $c_1 = 0$).*

$$\text{With } \lambda = \frac{(1-\tau_x)(\alpha+\mu)^2}{(L-\mu)n}, \quad \mathbb{E}_{i_k}[T_{k+1}] \leq \left(1 + \frac{\mu}{\alpha}\right)^{-2} T_k, \text{ for } k \geq 0.$$

Regrading the rate, from (14), we can figure out that $\alpha$ is the unique positive root of the cubic equation:

$$\left(\frac{\alpha}{\mu}\right)^3 - (2n - 3)\left(\frac{\alpha}{\mu}\right)^2 - (2n\kappa + n - 3)\left(\frac{\alpha}{\mu}\right) - (n\kappa - 1) = 0.$$

Using a similar argument as in Theorem 3, we can show that $\frac{\alpha}{\mu} = O(n + \sqrt{n\kappa})$, and thus conclude an $O\left((n + \sqrt{n\kappa})\log\frac{1}{\epsilon}\right)$ expected complexity for BS-SAGA. Interestingly, this rate is always slightly slower than that of BS-Point-SAGA.

### C.4.1 Proof of Theorem C.1

To simplify the notations in this proof, we let $\Phi^k \triangleq \frac{1}{n}\sum_{i=1}^n h_i(\phi_i^k)$ and $\nabla\Phi^k \triangleq \frac{1}{n}\sum_{i=1}^n \nabla h_i(\phi_i^k)$.

Using the trick in Lemma 3 (with $f = h_{i_k}$) for $\phi_{i_k}^{k+1}$, strengthening the convexity with the interpolation condition and taking the expectation, we obtain

$$
\begin{aligned}
\mathbb{E}_{i_k}\left[h_{i_k}(\phi_{i_k}^{k+1})\right] \leq{} & \frac{1-\tau_x}{\tau_x}\mathbb{E}_{i_k}\left[\langle\nabla h_{i_k}(\phi_{i_k}^{k+1}), \phi_{i_k}^k - \phi_{i_k}^{k+1}\rangle\right] + \mathbb{E}_{i_k}\left[\langle\nabla h_{i_k}(\phi_{i_k}^{k+1}), z_k - x^\star\rangle\right] \\
& + \frac{\tau_z}{\tau_x}\mathbb{E}_{i_k}\left[\left\langle\nabla h_{i_k}(\phi_{i_k}^{k+1}), \mu(\bar\phi^k - z_k) - \frac{1}{n}\sum_{i=1}^n\nabla f_i(\phi_i^k)\right\rangle\right] \\
& - \frac{1}{2(L-\mu)}\mathbb{E}_{i_k}\left[\|\nabla h_{i_k}(\phi_{i_k}^{k+1})\|^2\right].
\end{aligned}
$$

Note that by the definition of $h_i$, $\mu(\bar\phi^k - z_k) - \frac{1}{n}\sum_{i=1}^n\nabla f_i(\phi_i^k) = \mu(x^\star - z_k) - \nabla\Phi^k$, and thus

$$
\begin{aligned}
\mathbb{E}_{i_k}\left[h_{i_k}(\phi_{i_k}^{k+1})\right] \leq{} & \frac{1-\tau_x}{\tau_x}\mathbb{E}_{i_k}\left[\langle\nabla h_{i_k}(\phi_{i_k}^{k+1}), \phi_{i_k}^k - \phi_{i_k}^{k+1}\rangle\right] - \frac{\tau_z}{\tau_x}\mathbb{E}_{i_k}\left[\langle\nabla h_{i_k}(\phi_{i_k}^{k+1}), \nabla\Phi^k\rangle\right] \\
& + \left(1 - \frac{\mu\tau_z}{\tau_x}\right)\mathbb{E}_{i_k}\left[\langle\nabla h_{i_k}(\phi_{i_k}^{k+1}), z_k - x^\star\rangle\right] \\
& - \frac{1}{2(L-\mu)}\left\|\mathbb{E}_{i_k}\left[\nabla h_{i_k}(\phi_{i_k}^{k+1})\right]\right\|^2,
\end{aligned} \tag{15}
$$

which also uses Jensen's inequality, i.e., $\mathbb{E}_{i_k}\left[\|\nabla h_{i_k}(\phi_{i_k}^{k+1})\|^2\right] \geq \|\mathbb{E}_{i_k}\left[\nabla h_{i_k}(\phi_{i_k}^{k+1})\right]\|^2$.

Using Lemma 1 with $\mathcal{H}_y = \mathcal{H}_{\phi_{i_k}^{k+1}}^{\text{SAGA}}, \mathcal{G}_y = \mathcal{G}_{\phi_{i_k}^{k+1}}^{\text{SAGA}}, z^+ = z_{k+1}$ and taking the expectation, we obtain

$$
\begin{aligned}
\mathbb{E}_{i_k}\left[\langle\nabla h_{i_k}(\phi_{i_k}^{k+1}), z_k - x^\star\rangle\right] ={} & \frac{\alpha}{2}\left(\|z_k - x^\star\|^2 - \left(1 + \frac{\mu}{\alpha}\right)^2\mathbb{E}_{i_k}\left[\|z_{k+1} - x^\star\|^2\right]\right) \\
& + \frac{1}{2\alpha}\mathbb{E}_{i_k}\left[\left\|\mathcal{H}_{\phi_{i_k}^{k+1}}^{\text{SAGA}}\right\|^2\right].
\end{aligned} \tag{16}
$$

Using the interpolation condition of $h_{i_k}$ to bound the stochastic moment,

$$
\begin{aligned}
\mathbb{E}_{i_k}\left[\left\|\mathcal{H}_{\phi_{i_k}^{k+1}}^{\text{SAGA}}\right\|^2\right] ={} & \mathbb{E}_{i_k}\left[\|\nabla h_{i_k}(\phi_{i_k}^{k+1}) - \nabla h_{i_k}(\phi_{i_k}^k)\|^2\right] + 2\mathbb{E}_{i_k}\left[\langle\nabla h_{i_k}(\phi_{i_k}^{k+1}), \nabla\Phi^k\rangle\right] \\
& - \|\nabla\Phi^k\|^2 \\
\leq{} & 2(L-\mu)\left(\Phi^k - \mathbb{E}_{i_k}\left[h_{i_k}(\phi_{i_k}^{k+1})\right] - \mathbb{E}_{i_k}\left[\langle\nabla h_{i_k}(\phi_{i_k}^{k+1}), \phi_{i_k}^k - \phi_{i_k}^{k+1}\rangle\right]\right) \\
& + 2\mathbb{E}_{i_k}\left[\langle\nabla h_{i_k}(\phi_{i_k}^{k+1}), \nabla\Phi^k\rangle\right] - \|\nabla\Phi^k\|^2.
\end{aligned} \tag{17}
$$

Based on the updating rules of $\phi^{k+1}$, the following relations hold

$$
\mathbb{E}_{i_k}\left[\Phi^{k+1}\right] = \frac{1}{n}\mathbb{E}_{i_k}\left[h_{i_k}(\phi_{i_k}^{k+1})\right] + \frac{n-1}{n}\Phi^k, \tag{18}
$$

$$
\mathbb{E}_{i_k}\left[\nabla\Phi^{k+1}\right] = \frac{1}{n}\mathbb{E}_{i_k}\left[\nabla h_{i_k}(\phi_{i_k}^{k+1})\right] + \frac{n-1}{n}\nabla\Phi^k, \tag{19}
$$

where (19) implies that

$$
\begin{aligned}
\left\|\mathbb{E}_{i_k}\left[\nabla h_{i_k}(\phi_{i_k}^{k+1})\right]\right\|^2 ={} & n^2\|\mathbb{E}_{i_k}\left[\nabla\Phi^{k+1}\right]\|^2 - 2(n^2 - n)\langle\mathbb{E}_{i_k}\left[\nabla\Phi^{k+1}\right], \nabla\Phi^k\rangle \\
& + (n-1)^2\|\nabla\Phi^k\|^2,
\end{aligned} \tag{20}
$$

$$
\mathbb{E}_{i_k}\left[\langle\nabla h_{i_k}(\phi_{i_k}^{k+1}), \nabla\Phi^k\rangle\right] = n\langle\mathbb{E}_{i_k}\left[\nabla\Phi^{k+1}\right], \nabla\Phi^k\rangle - (n-1)\|\nabla\Phi^k\|^2. \tag{21}
$$

Then, expanding (15) using (16), (17), (20) and (21), we obtain

$$
\frac{1}{n}\mathbb{E}_{i_k}\left[h_{i_k}(\phi_{i_k}^{k+1})\right] \leq \left[\frac{1-\tau_x}{\tau_x n} - \left(1 - \frac{\mu\tau_z}{\tau_x}\right)\frac{L-\mu}{\alpha n}\right]\mathbb{E}_{i_k}\left[\langle\nabla h_{i_k}(\phi_{i_k}^{k+1}), \phi_{i_k}^k - \phi_{i_k}^{k+1}\rangle\right]
$$
$$
+ \left(1 - \frac{\mu\tau_z}{\tau_x}\right)\frac{L-\mu}{\alpha n}\left(\Phi^k - \mathbb{E}_{i_k}\left[h_{i_k}(\phi_{i_k}^{k+1})\right]\right)
$$
$$
+ \left(1 - \frac{\mu\tau_z}{\tau_x}\right)\frac{\alpha}{2n}\left(\|z_k - x^\star\|^2 - \left(1 + \frac{\mu}{\alpha}\right)^2\mathbb{E}_{i_k}\left[\|z_{k+1} - x^\star\|^2\right]\right)
$$
$$
+ \left[\frac{1}{\alpha} - \frac{\mu\tau_z}{\alpha\tau_x} - \frac{\tau_z}{\tau_x} + \frac{n-1}{L-\mu}\right]\langle\mathbb{E}_{i_k}\left[\nabla\Phi^{k+1}\right], \nabla\Phi^k\rangle
$$
$$
- \left[\frac{(n-1)^2}{2(L-\mu)n} + \left(1 - \frac{\mu\tau_z}{\tau_x}\right)\frac{1}{2\alpha n} + \left(\frac{1}{\alpha} - \frac{\mu\tau_z}{\alpha\tau_x} - \frac{\tau_z}{\tau_x}\right)\frac{n-1}{n}\right]\|\nabla\Phi^k\|^2
$$
$$
- \frac{n}{2(L-\mu)}\|\mathbb{E}_{i_k}\left[\nabla\Phi^{k+1}\right]\|^2.
$$

Choosing $\tau_z$ such that $\frac{1-\tau_x}{\tau_x} = \left(1 - \frac{\mu\tau_z}{\tau_x}\right)\frac{L-\mu}{\alpha}$, multiplying both sides by $\tau_x$ and using (18), we can simplify the above inequality as

$$
\mathbb{E}_{i_k}\left[\Phi^{k+1}\right] \leq \left(1 - \frac{\tau_x}{n}\right)\Phi^k + \frac{\alpha^2(1-\tau_x)}{2(L-\mu)n}\left(\|z_k - x^\star\|^2 - \left(1 + \frac{\mu}{\alpha}\right)^2\mathbb{E}_{i_k}\left[\|z_{k+1} - x^\star\|^2\right]\right)
$$
$$
+ \frac{\alpha + \mu - \tau_x(\alpha + L + \mu - \mu n)}{(L-\mu)\mu}\langle\mathbb{E}_{i_k}\left[\nabla\Phi^{k+1}\right], \nabla\Phi^k\rangle
$$
$$
- \frac{(n-2)\tau_x + \frac{1}{n} + \left(\frac{\alpha}{\mu} + 1 - \left(\frac{\alpha}{\mu} + \kappa\right)\tau_x\right)\left(2 - \frac{2}{n}\right)}{2(L-\mu)}\|\nabla\Phi^k\|^2
$$
$$
- \frac{n\tau_x}{2(L-\mu)}\|\mathbb{E}_{i_k}\left[\nabla\Phi^{k+1}\right]\|^2.
$$

Fixing $\tau_x = \frac{\alpha+\mu}{\alpha+L}$, we obtain

$$
\mathbb{E}_{i_k}\left[\Phi^{k+1}\right] \leq \left(1 - \frac{\tau_x}{n}\right)\Phi^k + \frac{\alpha^2(1-\tau_x)}{2(L-\mu)n}\left(\|z_k - x^\star\|^2 - \left(1 + \frac{\mu}{\alpha}\right)^2\mathbb{E}_{i_k}\left[\|z_{k+1} - x^\star\|^2\right]\right)
$$
$$
+ \frac{(n-1)\tau_x}{L-\mu}\langle\mathbb{E}_{i_k}\left[\nabla\Phi^{k+1}\right], \nabla\Phi^k\rangle - \frac{n\tau_x}{2(L-\mu)}\|\mathbb{E}_{i_k}\left[\nabla\Phi^{k+1}\right]\|^2
$$
$$
- \frac{(n-2)\tau_x + \frac{1}{n}}{2(L-\mu)}\|\nabla\Phi^k\|^2.
$$

Using Young's inequality with $\beta > 0$,

$$
\mathbb{E}_{i_k}\left[\Phi^{k+1}\right] \leq \left(1 - \frac{\tau_x}{n}\right)\Phi^k + \frac{\alpha^2(1-\tau_x)}{2(L-\mu)n}\left(\|z_k - x^\star\|^2 - \left(1 + \frac{\mu}{\alpha}\right)^2\mathbb{E}_{i_k}\left[\|z_{k+1} - x^\star\|^2\right]\right)
$$
$$
+ \frac{\beta(n-1)\tau_x - n\tau_x}{2(L-\mu)}\|\mathbb{E}_{i_k}\left[\nabla\Phi^{k+1}\right]\|^2 + \frac{\frac{(n-1)\tau_x}{\beta} - (n-2)\tau_x - \frac{1}{n}}{2(L-\mu)}\|\nabla\Phi^k\|^2.
$$

Let $\beta \in \left[\frac{n-1}{n-2+\frac{1}{n\tau_x}}, \frac{n}{n-1}\right]$. The last two terms become non-positive, and thus we have

$$
\mathbb{E}_{i_k}\left[\Phi^{k+1}\right] \leq \left(1 - \frac{\tau_x}{n}\right)\cdot\Phi^k + \frac{\alpha^2(1-\tau_x)}{2(L-\mu)n}\left(\|z_k - x^\star\|^2 - \left(1 + \frac{\mu}{\alpha}\right)^2\mathbb{E}_{i_k}\left[\|z_{k+1} - x^\star\|^2\right]\right).
$$

Letting $\left(1 - \frac{\tau_x}{n}\right)\left(1 + \frac{\mu}{\alpha}\right)^2 = 1$ completes the proof.

## D Proof for Section 5 (Theorem 3)

Using Lemma 2 with the relations

$$x_{k+1} = \text{prox}_{i_k}^{\alpha}\left(x_k + \frac{1}{\alpha}\left(\nabla f_{i_k}(\phi_{i_k}^k) - \frac{1}{n}\sum_{i=1}^n \nabla f_i(\phi_i^k) + \mu\left(\frac{1}{n}\sum_{i=1}^n \phi_i^k - \phi_{i_k}^k\right)\right)\right),$$

$$x^\star = \text{prox}_{i_k}^{\alpha}\left(x^\star + \frac{1}{\alpha}\nabla f_{i_k}(x^\star)\right) \text{ and } \phi_{i_k}^{k+1} = x_{k+1},$$

and based on that $\nabla h_i(x) = \nabla f_i(x) - \nabla f_i(x^\star) - \mu(x - x^\star)$, we have

$$\left(1 + \frac{2(\alpha + \mu)}{L - \mu}\right)\|\nabla h_{i_k}(\phi_{i_k}^{k+1})\|^2 + (\alpha + \mu)^2\|x_{k+1} - x^\star\|^2$$

$$\leq \alpha^2\left\|x_k - x^\star + \frac{1}{\alpha}\left(\nabla h_{i_k}(\phi_{i_k}^k) - \frac{1}{n}\sum_{i=1}^n \nabla h_i(\phi_i^k)\right)\right\|^2.$$

Expanding the right side, taking the expectation and using $\mathbb{E}[\|X - \mathbb{E}X\|^2] \leq \mathbb{E}[\|X\|^2]$, we obtain

$$\left(1 + \frac{2(\alpha + \mu)}{L - \mu}\right)\mathbb{E}_{i_k}\left[\|\nabla h_{i_k}(\phi_{i_k}^{k+1})\|^2\right] + (\alpha + \mu)^2\mathbb{E}_{i_k}\left[\|x_{k+1} - x^\star\|^2\right]$$

$$\leq \alpha^2\|x_k - x^\star\|^2 + \frac{1}{n}\sum_{i=1}^n \|\nabla h_i(\phi_i^k)\|^2.$$

Note that by construction,

$$\mathbb{E}_{i_k}\left[\sum_{i=1}^n \|\nabla h_i(\phi_i^{k+1})\|^2\right] = \frac{n-1}{n}\sum_{i=1}^n \|\nabla h_i(\phi_i^k)\|^2 + \mathbb{E}_{i_k}\left[\|\nabla h_{i_k}(\phi_{i_k}^{k+1})\|^2\right].$$

We can thus arrange the terms as

$$\left(\frac{n}{\alpha^2} + \frac{2(\alpha + \mu)n}{\alpha^2(L - \mu)}\right)\mathbb{E}_{i_k}\left[\frac{1}{n}\sum_{i=1}^n \|\nabla h_i(\phi_i^{k+1})\|^2\right] + \left(1 + \frac{\mu}{\alpha}\right)^2\mathbb{E}_{i_k}\left[\|x_{k+1} - x^\star\|^2\right]$$

$$\leq \left(\frac{n}{\alpha^2} + \frac{2(\alpha + \mu)(n-1)}{\alpha^2(L - \mu)}\right)\cdot\frac{1}{n}\sum_{i=1}^n \|\nabla h_i(\phi_i^k)\|^2 + \|x_k - x^\star\|^2.$$

In view of the Lyapunov function, we choose $\alpha$ to be the positive root of the following equation:

$$\left(1 + \frac{\mu}{\alpha}\right)^2\left(1 - \frac{2(\alpha + \mu)}{n(L - \mu) + 2n(\alpha + \mu)}\right) = 1.$$

Let $q = \frac{\alpha}{\mu} > 0$, the above is a cubic equation:

$$s(q) \triangleq 2q^3 - (4n - 6)q^2 - (2n\kappa + 4n - 6)q - (n\kappa + n - 2) = 0,$$

which has a unique positive root (denoted as $q^\star$).

Note that $s(-\infty) < 0, s(-\frac{1}{2}) = \frac{1}{4}$ and $s(0) \leq 0$. These facts suggest that if for some $u > 0$, $s(u) > 0$, we have $q^\star < u$. It can be verified that $s(2n + \sqrt{n\kappa}) > 0$, and thus $q^\star = O(n + \sqrt{n\kappa})$.

## E Experimental setup

We ran experiments on an HP Z440 machine with a single Intel Xeon E5-1630v4 with 3.70GHz cores, 16GB RAM, Ubuntu 18.04 LTS with GCC 4.8.0, MATLAB R2017b. We were optimizing the

following binary problems with $a_i \in \mathbb{R}^d$, $b_i \in \{-1, +1\}$, $i \in [n]$:

$$\ell_2\text{-Logistic Regression: } \frac{1}{n} \sum_{i=1}^{n} \log \left(1 + \exp\left(-b_i \langle a_i, x \rangle\right)\right) + \frac{\mu}{2} \|x\|^2,$$

$$\text{Ridge Regression: } \frac{1}{2n} \sum_{i=1}^{n} \left(\langle a_i, x \rangle - b_i\right)^2 + \frac{\mu}{2} \|x\|^2.$$

We used datasets from the LIBSVM website [9], including a9a (32,561 samples, 123 features), covtype.binary (581,012 samples, 54 features), w8a (49,749 samples, 300 features), ijcnn1 (49,990 samples, 22 features). We added one dimension as bias to all the datasets.

We choose SAGA and Katyusha as the baselines in the finite-sum experiments due to the following reasons: SAGA has low iteration cost and good empirical performance with support for non-smooth regularizers, and is thus implemented in machine learning libraries such as scikit-learn [39]; Katyusha achieves the state-of-the-art performance for ill-conditioned problems[11].

# F Analyzing NAG using Lyapunov function

---

**Algorithm 5** Nesterov's Accelerated Gradient (NAG)

---

**Input:** Parameters $\alpha > 0, \tau_y, \tau_x \in ]0, 1[$ and initial guesses $x_0, z_0 \in \mathbb{R}^d$, iteration number $K$.
 1: **for** $k = 0, \dots, K - 1$ **do**
 2: $\quad y_k = \tau_y z_k + (1 - \tau_y) x_k$.
 3: $\quad z_{k+1} = \arg\min_x \left\{ \langle \nabla f(y_k), x \rangle + (\alpha/2)\|x - z_k\|^2 + (\mu/2)\|x - y_k\|^2 \right\}$.
 4: $\quad x_{k+1} = \tau_x z_{k+1} + (1 - \tau_x) x_k$.
 5: **end for**
**Output:** $x_K$.

---

In this section, we review the convergence of NAG in the strongly convex setting for a better comparison with the convergence guarantee and proof of G-TM. This Lyapunov analysis has been similarly presented in many existing works, e.g., [54, 18, 5, 38]. We adopt a simplified version of NAG in Algorithm 5 (1-memory accelerated methods, [52]) and only consider constant parameter choices. It is known that NAG can be analyzed based on the following Lyapunov function ($\lambda > 0$):

$$T_k = f(x_k) - f(x^\star) + \frac{\lambda}{2}\|z_k - x^\star\|^2, \tag{22}$$

which is somehow suggested in the construction of the *estimate sequence* in [35]. This choice requires neither $f(x_k) - f(x^\star)$ nor $\|z_k - x^\star\|^2$ to be monotone decreasing over iterations, which is called the non-relaxational property in [32]. By re-organizing the proof in [35] under the notion of Lyapunov function, we obtain the per-iteration contraction of NAG in Theorem F.1.

**Theorem F.1.** *In Algorithm 5, suppose we choose $\alpha, \tau_x, \tau_y$ under the constraints* (23)*, the iterations satisfy the contraction* (24) *for the Lyapunov function* (22)*.*

$$\begin{cases} \alpha \geq \frac{L(1-\tau_x)\tau_y}{1-\tau_y}, \tau_x \geq \tau_y, \\ \mu \geq \frac{L(\tau_x - \tau_y)}{1 - \tau_y}, \\ \left(1 + \frac{\mu}{\alpha}\right)(1 - \tau_x) \leq 1. \end{cases} \tag{23} \qquad \begin{aligned} &\textit{With } \lambda = (\alpha + \mu)\tau_x, \\ &T_{k+1} \leq \left(1 + \frac{\mu}{\alpha}\right)^{-1} T_k, \textit{ for } k \geq 0. \end{aligned} \tag{24}$$

When the inequalities in constraints (23) (except $\tau_x \geq \tau_y$) hold as equality, we derive the standard choice of NAG: $\alpha = \sqrt{L\mu} - \mu, \tau_y = (\sqrt{\kappa} + 1)^{-1}, \tau_x = (\sqrt{\kappa})^{-1}$. By substituting this choice and

eliminating sequence $\{z_k\}$, we recover the widely-used scheme (Constant Step scheme III in [35]):

$$x_{k+1} = y_k - \frac{1}{L}\nabla f(y_k),$$

$$y_{k+1} = x_{k+1} + \frac{\sqrt{\kappa}-1}{\sqrt{\kappa}+1}(x_{k+1} - x_k).$$

Telescoping (24), we obtain the original guarantee of NAG (cf. Theorem 2.2.3 in [35]),

$$f(x_K) - f(x^\star) + \frac{\mu}{2}\|z_K - x^\star\|^2 \leq \left(1 - \frac{1}{\sqrt{\kappa}}\right)^K \left(f(x_0) - f(x^\star) + \frac{\mu}{2}\|z_0 - x^\star\|^2\right).$$

If we regard the constraints (23) as an optimization problem with a target of minimizing the rate factor $(1 + \frac{\mu}{\alpha})^{-1}$, the rate factor $1 - 1/\sqrt{\kappa}$ is optimal. Combining $\alpha \geq \frac{L(1-\tau_x)\tau_y}{1-\tau_y}$ and $\mu \geq \frac{L(\tau_x - \tau_y)}{1-\tau_y}$, we have $\alpha \geq L\tau_x - \mu$. To minimize $\alpha$, we fix $\alpha = L\tau_x - \mu$, and it can be easily verified that in this case, the smallest rate factor is achieved when $\left(1 + \frac{\mu}{\alpha}\right)(1 - \tau_x) = 1$. Note that these arguments do not consider variable-parameter choices and are limited to the current analysis framework only.

Denote the initial constant as $C_0^{\text{NAG}} \triangleq f(x_0) - f(x^\star) + \frac{\mu}{2}\|z_0 - x^\star\|^2$. This guarantee shows that in terms of reducing $\|x - x^\star\|^2$ to $\epsilon$, sequences $\{x_k\}$ and $\{z_k\}$ have the same iteration complexity $\sqrt{\kappa} \log \frac{2C_0^{\text{NAG}}}{\mu\epsilon}$. Since $\{y_k\}$ is a convex combination of them, it also converges with the same complexity.

### F.1 Proof of Theorem F.1

For the convex combination $y_k = \tau_y z_k + (1 - \tau_y) x_k$, we can use the trick in Lemma 3 to obtain

$$
\begin{aligned}
f(y_k) - f(x^\star) &\leq \frac{1-\tau_y}{\tau_y}\langle \nabla f(y_k), x_k - y_k\rangle + \langle \nabla f(y_k), z_k - x^\star\rangle - \frac{\mu}{2}\|y_k - x^\star\|^2 \\
&= \frac{1-\tau_y}{\tau_y}\langle \nabla f(y_k), x_k - y_k\rangle + \underbrace{\langle \nabla f(y_k), z_k - z_{k+1}\rangle}_{R_1} \\
&\quad + \underbrace{\langle \nabla f(y_k), z_{k+1} - x^\star\rangle}_{R_2} - \frac{\mu}{2}\|y_k - x^\star\|^2.
\end{aligned}
\tag{25}
$$

For $R_1$, based on the $L$-smoothness, we have

$$f(x_{k+1}) - f(y_k) + \langle \nabla f(y_k), y_k - x_{k+1}\rangle \leq \frac{L}{2}\|x_{k+1} - y_k\|^2.$$

Note that $y_k - x_{k+1} = \tau_x(z_k - z_{k+1}) + (\tau_y - \tau_x)(z_k - x_k)$, we can arrange the above inequality as

$$f(x_{k+1}) - f(y_k) + \langle \nabla f(y_k), \tau_x(z_k - z_{k+1}) + (\tau_y - \tau_x)(z_k - x_k)\rangle \leq \frac{L}{2}\|x_{k+1} - y_k\|^2,$$

$$R_1 \leq \frac{L}{2\tau_x}\|x_{k+1} - y_k\|^2 + \frac{1}{\tau_x}\left(f(y_k) - f(x_{k+1})\right) - \frac{\tau_y - \tau_x}{\tau_x}\langle \nabla f(y_k), z_k - x_k\rangle. \tag{26}$$

For $R_2$, based on the optimality condition of the 3rd step in Algorithm 5, which is for any $u \in \mathbb{R}^d$,

$$\langle \nabla f(y_k) + \alpha(z_{k+1} - z_k) + \mu(z_{k+1} - y_k), u - z_{k+1}\rangle = 0,$$

we have (by choosing $u = x^\star$),

$$
\begin{aligned}
R_2 &= \alpha\langle z_{k+1} - z_k, x^\star - z_{k+1}\rangle + \mu\langle z_{k+1} - y_k, x^\star - z_{k+1}\rangle \\
&= \frac{\alpha}{2}\left(\|z_k - x^\star\|^2 - \|z_{k+1} - x^\star\|^2 - \|z_{k+1} - z_k\|^2\right) \\
&\quad + \frac{\mu}{2}\left(\|y_k - x^\star\|^2 - \|z_{k+1} - x^\star\|^2 - \|z_{k+1} - y_k\|^2\right).
\end{aligned}
\tag{27}
$$

By upper bounding (25) using (26), (27), we can conclude that

$$f(y_k) - f(x^\star) \le \frac{1 - \tau_x}{\tau_x} \langle \nabla f(y_k), x_k - y_k \rangle + \frac{1}{\tau_x} \big( f(y_k) - f(x_{k+1}) \big)$$
$$+ \frac{\alpha}{2} \left( \|z_k - x^\star\|^2 - \left(1 + \frac{\mu}{\alpha}\right) \|z_{k+1} - x^\star\|^2 \right)$$
$$+ \frac{L}{2\tau_x} \|x_{k+1} - y_k\|^2 - \frac{\alpha}{2} \|z_{k+1} - z_k\|^2 - \frac{\mu}{2} \|z_{k+1} - y_k\|^2,$$

Re-arrange the terms,

$$f(x_{k+1}) - f(x^\star) \le (1 - \tau_x)\big(f(x_k) - f(x^\star)\big) + \frac{\alpha \tau_x}{2} \left( \|z_k - x^\star\|^2 - \left(1 + \frac{\mu}{\alpha}\right) \|z_{k+1} - x^\star\|^2 \right)$$
$$+ \frac{L}{2} \|x_{k+1} - y_k\|^2 - \frac{\alpha \tau_x}{2} \|z_{k+1} - z_k\|^2 - \frac{\mu \tau_x}{2} \|z_{k+1} - y_k\|^2. \tag{28}$$

Note that the following relation holds:

$$x_{k+1} - y_k = \tau_x \left( \frac{(1 - \tau_x)\tau_y}{(1 - \tau_y)\tau_x}(z_{k+1} - z_k) + \frac{\tau_x - \tau_y}{(1 - \tau_y)\tau_x}(z_{k+1} - y_k) \right),$$

and thus if $\tau_x \ge \tau_y$, based on the convexity of $\|\cdot\|^2$, we have

$$\frac{L}{2} \|x_{k+1} - y_k\|^2 \le \frac{L(1 - \tau_x)\tau_x \tau_y}{2(1 - \tau_y)} \|z_{k+1} - z_k\|^2 + \frac{L(\tau_x - \tau_y)\tau_x}{2(1 - \tau_y)} \|z_{k+1} - y_k\|^2.$$

Finally, suppose that the following relations hold

$$\begin{cases} \tau_x \ge \tau_y, \\ \alpha \ge \frac{L(1 - \tau_x)\tau_y}{1 - \tau_y}, \\ \mu \ge \frac{L(\tau_x - \tau_y)}{1 - \tau_y}, \\ \left(1 + \frac{\mu}{\alpha}\right)(1 - \tau_x) \le 1, \end{cases}$$

we can arrange (28) as

$$f(x_{k+1}) - f(x^\star) + \frac{\alpha \tau_x}{2} \left(1 + \frac{\mu}{\alpha}\right) \|z_{k+1} - x^\star\|^2$$
$$\le \left(1 + \frac{\mu}{\alpha}\right)^{-1} \left( f(x_k) - f(x^\star) + \frac{\alpha \tau_x}{2} \left(1 + \frac{\mu}{\alpha}\right) \|z_k - x^\star\|^2 \right),$$

which completes the proof.

## Footnotes

[11]Zhou et al. [58] shows that SSNM can be faster than Katyusha in some cases. In theory, SSNM and Katyusha achieve the same rate if we set $m = n$ for Katyusha (both require 2 oracle calls per-iteration). In practice, if $m = n$, they have similar performance (SSNM is often faster). Considering the stability and memory requirement, Katyusha still achieves the state-of-the-art performance both theoretically and empirically.


[Supplementary Material 2 · README.pdf]

# Demo for BS-SVRG

A demo for SVRG Boosted by Shifting Objective (BS-SVRG) proposed in "Boosting First-order Methods by Shifting Objective: New Schemes with Faster Worst Case Rates".

## Usage

All algorithms are implemented in C++.

To run the demo in MATLAB, first run `mex_all` in the MATLAB terminal to generate the mex file. (Note that the compiler should support at least `c++11`)

Then, run `TEST` in the MATLAB terminal, a small demo training $\ell2$-logistic regression $(\mu = 5 \times 10^{-8})$ using dataset `a9a` from LIBSVM Data, to generate a plot shown as below.

Test environment: HP Z440 machine with single Intel Xeon E5-1630v4 with 3.70GHz cores, 16GB RAM, Ubuntu 18.04 LTS with GCC 4.8.0, MATLAB R2017b.

```
>> TEST
Building with 'g++'.
MEX completed successfully.
Model: L2-logistic
Algorithm: SAGA
Time: 11.582018 seconds
Algorithm: Katyusha
Time: 15.778917 seconds
Algorithm: BS_SVRG
Time: 8.186314 seconds
Algorithm: BS_SVRG
Time: 8.189970 seconds
```