[Reviews · NeurIPS 2020]

Review 1

Summary and Contributions: ********* After rebuttal ********* I thank the authors for their answers and wish to raise my score to "accept" (as announced to the AC, conditioned by the rebuttal). Let me add a few comments. Please, again, don't see my comments below as an incentive to add references in case they are not justified by the changes made by the authors. (1) About the shape of the proofs: yes, but for me the "[...] and eliminate residual terms." is exactly what makes the proof "algebraic". Beyond that, I agree with the authors. (2) Concerning the references I provided, I let the author judge which one are relevant: a literature check might be welcome here, as many persons came up with similar Lyapunov functions, although the idea is quite old. (to illustrate my claim: in the non-strongly convex case, the Lyapunov function of type a_k (f(x_k)-f_*) + L/2 ||z_k-x_*||^2 is already to be found in Nesterov's 1983 paper, as an intermediate inequality). (3) Concerning the non-strongly convex case, I wish to add that the Lyapunov function provided by the authors was already used in at least one previous work for obtaining an accelerated method, in a similar line of work as that of Drori & Teboulle, Kim & Fessler, Van Scoy et al., and Lessard et al.: - Stochastic first-order methods: non-asymptotic and computer-aided analyses via potential functions (Taylor & Bach, 2019), Theorem 11. This shows that adding relevant bibliographical entries (probably not only this one?) might be necessary, would the authors consider mentioning the non-strongly convex case. (4) Another remark, concerning the proximal version. One first possibility could be to assume the smooth convex term to be defined everywhere (just as FISTA does). If I am not mistaken it was attempted in (the "proximal optimized gradient method"; but there was no proof, only numerics): - Exact Worst-case Performance of First-order Methods for Composite Convex Optimization (Taylor, Hendrickx, Glineur, 2018), - Adaptive restart of the optimized gradient method for convex minimization (Kim & Fessler, 2018). Another possibility would be to adapt the inequalities, such as in - Another look at the fast iterative shrinkage/thresholding algorithm (FISTA) (Kim & Fessler, 2018). ************************************ This paper presents a new idea for developing faster first-order methods. Put in my humble words, the idea is to change the typical Lyapunov functions (using what the authors call "cocoercivity" (which is not the right terminology, hence the quotation marks)) used in worst-case analyses. The idea is based on recent works [7] and [40], in which the Lyapunov function presented for analyzing the "Triple Momentum Method" has a non-standard shape. The authors successfully apply this idea to develop new stochastic methods: BS-SVRG and BS-SAGA. They also propose a framework called Generalized Triple Momentum (G-TM), and an alternative to Point-SAGA for when proximal operators are available. Overall, the stochastic methods presented in this work improve upon the known rates for finite sum problems.

Strengths: Overall, I liked the idea, and see the following strengths in the paper: - the proofs and rates of the new stochastic methods are indeed improvements over known methods and results. - The idea of using "cocoercivity" in the Lyapunov function (or the "shift" as termed by the author(s)) is valuable and can certainly be digged further in other settings. I believe it is of interest for the community, as (i) the rates are indeed improved, (ii) understanding "base optimization schemes/settings" is certainly a pre-requisite for understanding optimization routines in more complex settings. Overall, I believe this work makes a step in the direction of getting cleaner results in this area. A number of the works that were improved here were accepted in previous editions of Neurips/Nips.

Weaknesses: On the other side, I like the topic and I believe the authors were not totally careful in the treatment of several aspects. - I would appreciate if the authors could provide a table for comparing algorithmic parameters of NAG, TMM, and G-TM. I currently have the impression that some of the claimed problems of the Triple Momentum Method are a bit artificial. Whereas G-TM is indeed a generalization, the optimal parameter choices (made by the authors) essentially (up to the initialization) correspond to those of the Triple Momentum method in [7] and [40], if I am not mistaken. - The algorithms presented by the authors rely on what they call "cocoercivity" (again, this is not the good wording, see comments below), making the methods hardly generalizable to handle constraints/proximal terms (because this condition requires the smooth functions to be defined everywhere). In other words, those inequalities actually heavily exploit the particular fact that the problem is unconstrained. - Honestly, I still believe the proofs to be purely algebraic (at least, I don't fully understand the intuition beyond making the algebra work), and mostly intuitive for their authors, although the authors claim they are less algebraic than that of TMM (the proof in [7] is actually, imho, pretty simple). In short, if the submission was to a journal, I would encourage some modifications before acceptation.

Correctness: From what I verified (I did not go through all detailed developments in the appendix), everything seemed fine.

Clarity: The text is well written and essentially easy to follow. I also did not find many typos (but I also did not hunt them); see comments below. However, I have to say I strongly believe I did not fully understand the idea in the way the author(s) wanted me to, as everything remained quite algebraic to me.

Relation to Prior Work: Overall, I believe it is the case. However, I would like the authors to explicit the difference between G-TM and TMM in the core of the paper, e.g., by explicitly providing the corresponding guarantee for TMM (Equation (11) in [7]) and a table for comparing parameter choices for NAG, TMM and G-TM (optimal tuning).

Reproducibility: Yes

Additional Feedback: I gave some references in the following questions & comments, please don't take it in any way as an incentive to cite any of those papers. I just put them to possibly clarify my statements. Questions & comments: - [about comparison with TMM] (i) I humbly believe the "redundant parameters" claimed for the triple momentum method is somewhat artificial. In a way, one of the point of [7] is that the only required parameter is rho. In addition, by picking rho=1-1/sqrt(kappa) as prescribed by TMM, there is no free parameter. So I would humbly replace this statement by something saying that the authors, on the contrary, allow picking more general values of the parameters. (ii) Also, about the flawed guarantee: it seems to me that the + log \kappa term is due to the leading condition ratio in Equation (4) in [40, optimization online version]; I guess the reason for this leading term is indeed that given by the authors; but this intermediate inequality might help understanding this. (iii) About the highly algebraic proof: imho, the Lyapunov analysis from [7] is rather simple, as it only requires combining two inequalities. (iv) I would appreciate a table instantiating the algorithmic parameters of G-TM for the different schemes. - [cocoercivity] A detail, but I think it is important to stick to the community language: what the author(s) call "cocoercivity" ( f(x) >= f(y) + <f'(y);x-y> + 1/2/L * || f'(x)-f'(y)||^2 ) is a consequence of smoothness, but is not called cocoercivity. Cocoercity is a consequence of this inequality, not involving function values: <f'(x)-f(y);x-y> >= 1/L * || f'(x)-f'(y)||^2. This is important, as for example, smooth convex functions have cocoercive gradients, but the inequality involving function values is only valid for smooth convex functions defined everywhere on R^d (see "On the properties of convex functions over open sets" by Y. Drori), whereas cocoercivity of the gradient can be used for characterizing smooth functions that are only defined on a subset of R^d (see for example "Worst-case convergence analysis of gradient and Newton methods through semidefinite programming performance estimation" by de Klerk et al.) - [about "distinctive features"] A few details: (i) lines 28-29: "and is very tight": what do the authors mean? (ii) Related to that: lines 29-31 & lines 36-37: "this inequality is the only property we need": it is shown in your reference [38] that any convergence result of the type the authors are proving, for unconstrained smooth (possibly strongly) convex minimization is a consequence of this inequality (via "smooth strongly convex interpolation"). Therefore, it is not a feature of the approach of the authors to use only this inequality, it is a feature of unconstrained smooth (strongly) convex optimization. (Btw, I guess their techniques could allow checking tightness results the authors were looking for). - In appendix F: just to be clear, there are already many Lyapunov analyses of Nesterov's method. The first one was implicitly used (it is in the middle of the equations) in the first acceleration paper of Nesterov ("A method for solving the convex programming problem with convergence rate O (1/k^2)"). Other ones are presented e.g., in "Dissipativity theory for Nesterov's accelerated method" (Hu & Lessard), "Potential-function proofs for first-order methods" (Bansal & Gupta), "Potential-based analyses of first-order methods for constrained and composite optimization" (Paquette & Vavasis), and many others. Typos/minor comments: - footnote 1: "but probably" -> "typically" ? - line 30: "the only property we need" -> specify that you need it for all h_i with i\in [n] ? - lines 135-139: y_{-1} and z_0 could be picked such that y_1 = y_0, if two evaluations of a gradient for the first iteration is too much for the author(s). - line 201: missing a second closing bracket before "contraction". - line 153: "monotone" is usually implicitly used for decreasing function values. I am not sure this is a problem, though. - line 170: thought experiment: -> no "." instead of ":"? (this appeared at different places) - lines 171-180: this thought experiment is not that clear for me. - lines 235-238: no capital letters after "(1)", "(2)", and "(3)"? - lines 261 and 269: the word "interesting" is subjective and should probably be avoided. - line 265: another possibility would be that worst-case analyses might not be representative of reality for such settings/algorithms (if e.g., worst-case scenarios are not stable to perturbations). - Appendix E, footnote 8: "slightly" faster -> I would suggest changing it to "typically" or "often" as I am unsure it is possible to prove it is always faster.


Review 2

Summary and Contributions: The paper proposes a novel strategy to develop algorithms and their convergence rates via shifting the optimization finite sum objective. I like the idea a lot -- the shifted objective appears more natural, which is confirmed by slightly tighter rates (constant times) and simpler proofs on multiple examples such as AGD, triple momentum, SVRG, SAGA, and point SAGA. I am surprised that I have never seen such an approach done earlier; good job! ################### After rebuttal ################### I am also satisfied with the author feedback; thanks a lot! I believe that now I understand the methodology behind developing new methods/proofs via "shifted objective. Unfortunately, the paper and the answer still do not provide a bigger picture on how the shifted objective can help the (first order) optimization algorithms in general besides the several examples provided in the paper. Overall, I believe that the paper can be significantly improved if the results were "sold" differently -- I would rate the paper higher if that was done. However, I believe that the paper is still worth publishing even in the current form; I like a lot the presented ideas and I agree with authors that the analysis/new algorithms are more natural. Stressing again what I wrote in the original review, I am surprised that the main "trick" of the paper was not done yet given how simple and natural it is. I can imagine rating this paper 8 if it was written differently and if the focus of the paper was slightly different. For the reasons above, I decided to keep my score (6). Nice work, it was my pleasure to review it!

Strengths: The paper is very relevant to Neurips/ML community. The contribution is quite significant as the main idea of the paper is very simple, but can be very influential at the same time. The results are theoretically grounded (I did some check of the proofs, not all though).

Weaknesses: There are, however, several issues that need to be addressed, mostly from the presentation side. First and foremost, the abstract promises an "algorithmic template for shifted objective". However, at the current form, the paper looks like a random collection of methods; while it is not directly mentioned how exactly these were derived from the shifting theory. I presume that BS-SVRG and BS-SAGA are obtained as (accelerated) SVRG/SAGA applied on the shifted objective; is that the case? If so, please dedicate at least a paragraph in each section to properly explain this (includuing BS-SVRG where already the shifted gradient estimator is mentioned) The best example for this is Section 3.1. The G-TM algorithm is not motivated enough (in fact, it is not motivated at all). It is not clear whether G-TM was derived from the shifted objective (I hope so; in such a case it would be good to see how exactly) or whether G-TM is just a random method that does not fit in the paper. In general, I encourage the authors to take a step back, focus a bit less on the proof techniques and rather properly motivate how the methods were obtained at the first place. For example, it would be great to see a detailed discussion on what are the limitations of the methodology; i.e., on top of which methods it can be applied (even without a proof). For example, can one develop BS-SGD? How about randomized coordinate descent? How about proximal methods with extra regularizer? I presume that one limitation of such an approach would be the prior need for knowing the strongly convexity constant (no difference for acceleration) Minor: line 51: start the sentence with a word Lemma 1: mention that G is a gradient estimator, not just the estimator (or alternatively, mention the space of G)

Correctness: I did some check of the proofs (not all though), and all seems correct.

Clarity: The writing can be improved (see weaknesses)

Relation to Prior Work: yes

Reproducibility: Yes

Additional Feedback: Overall, I like the idea of the paper a lot and I believe it deserves a publication. On the other hand, I believe that the presentation of the paper can be significantly improved. I am happy to raise my score given that the concerns I raised get fixed.


Review 3

Summary and Contributions: The paper studies first-order methods for unconstrained strongly convex problems. They provide an algorithm template and schemes that handle the shifted objective function in a co-coercivity condition (which gives tighter analysis as they've shown in the paper). It also works for problems with a variety of first-order oracles.

Strengths: The highlights of this work are various methods invented to solve a shifted objective.

Weaknesses: Experiments conducted on machine learning problems (and data sets) in this paper are not comprehensive enough. For example, they could add more commonly used machine learning datasets and baseline methods.

Correctness: The experimental methodology is correct.

Clarity: The paper is very well written.

Relation to Prior Work: The related work section is comprehensive.

Reproducibility: Yes

Additional Feedback:


Review 4

Summary and Contributions: The paper introduces a new shifted objective for a large class of classical optimization problems, which allow the use of tighter inequalities, and thus better rates for accelerated variance reduced stochastic gradient descent algorithms such as SAGA and SVRG. A wealth of theoretical results are also provided for the strongly convex case, demonstrating improved rates compared to the previous methods which do not use the shifted objective. Experimental results provide further insights in the applicability of the proposed methods.

Strengths: The derived rates are better than the ones in the previous literature, while using a somewhat simpler update scheme, which is a significant novel contribution. A lot of different problems are studied (n=1, finite sum, proximal setting), and several variants are proposed, based on SVRG, SAGA and TM, showcasing the applicability of the co-coercivity approach.

Weaknesses: While there is a laudable effort to illustrate the formulas in the theoretical results, these remain hard to parse. One imagines it would have been possible to give slightly less tight formulations that would have made the understanding easier. In general the paper feels a bit cramped, with a lot of material covered in just 8 pages. Another issue regarding significance is that these results are only interesting in the strongly convex setting, which somewhat limits their applicability (it's still an interesting problem to work on, but sadly will not directly benefit a lot of ML practitioners these days).

Correctness: The claims seem reasonable and the experiments complement the theoretical aspects well. I have not delved deeply in the mathematical proofs.

Clarity: The paper is generally well written, although very compact. For ease of reading, spacing things out more would have been more effective.

Relation to Prior Work: The prior work is discussed at length both in an isolated part as well as throughout the papers for relevant comparisons/contrasts, which is very appreciated. The delineation between the contributions and the previous work is very well done.

Reproducibility: Yes

Additional Feedback: Update after rebuttal: Having read all reviews and the authors' response, I encourage the authors to take the feedback into account in terms of presentation, as it could improve the reader experience (and overall quality of the paper) significantly.

[Author Response · NeurIPS 2020]

We thank the reviewers for taking the time to carefully read the paper and their constructive comments. We see a
common concern on the lack of discussion on the limitations/possible extensions of our methods, which we discuss
below and will give more details in the paper:
(1) Proximal/constrained setting: As pointed out by Reviewer#1, currently, the proposed methods do not work in this
setting as they rely on the "co-coercivity" property (we will correct the terminology). We may consider dropping
the negative gradient norms in their Lyapunov functions, which leads to less tight formulations and slower rates (as
suggested by Reviewer#4) but also makes them less reliant on that property. We think this might be feasible.
(2) Prior knowledge of strong convexity constant $\mu$: This methodology requires a known $\mu$ since even if it is applied to
a non-accelerated method, the parameter choice is always related to $\mu$.
(3) Non-strongly convex case: After submission, we discovered that when $\mu = 0$ in the framework of G-TM, it is
possible to adopt a variable parameter setting (the benefit of allowing variable choices) that leads to the $O(1/K^2)$ rate.
The special part is that, since the Lyapunov function becomes $T_k = a_k \cdot \left( f(y_k) - f(x^\star) - \frac{1}{2L}\|\nabla f(y_k)\|^2 \right) + \frac{L}{4}\|z_k - x^\star\|^2$,
one step GD at $y_{K-1}$ is critical to obtain a convergence guarantee. We suspect that this scheme is equivalent to the
optimized gradient method (Kim and Fessler, 2016), which could answer some of the questions raised in that work.
(4) BS-SGD is a promising direction, which requires considerable efforts (Moulines and Bach, 2011).

**To Reviewer#1** Thank you for your detailed comments. Please also see the revision plan to Reviewer#2.
[Comparison with TMM] We admit that the claimed "redundant parameters" problem of TMM is a bit artificial and
will make the following two changes to improve the comparison: (i) we will add a table comparing the parameters of
NAG, TMM and G-TM (optimal tuning), and provide the guarantee of TMM (Eq.(11) in [7]) in Section 3.1; (ii) we will
change "redundant parameters" to "a general scheme" that allows variable parameter choices and is easier to extend
(see extension (3) above). About the flawed guarantee, thanks for pointing out the intermediate inequality. Actually, in
our derivation of the original TMM (we started from Eq.(11) in [7]), it is possible to obtain an initial constant that is
roughly equal to that of NAG (i.e., $C_0^{\mathrm{NAG}}$) when $\kappa$ is large. This can also be seen in Figure 2a where TMM hits the
upper bound of NAG in the first iteration. We didn't discuss this because we thought G-TM already resolves this issue.
[About the proofs] For a Lyapunov function $T_k = a_k h(y_k) + b_k\|z_k - x^\star\|^2$, our strategy is to first build contractions
for $h(y_k)$ (Lemma 3) and $\|z_k - x^\star\|^2$ (Lemma 1), and then sum them up and eliminate residual terms. Comparing with
the proof in [7], the key difference is that we point out Lemma 1, which allows shifted stochastic gradient and reads as a
classic inequality whose usage has been well studied. We may instead say our proofs are more extensible.
[Co-coercivity] Thanks for the detailed clarification. We will revise related sentences following existing literature.
[Appendix F] Thank you for suggesting the existing works. We will mention that Appendix F is just for completeness.
[Typos/minor comments] We will fix the typos and try to improve the unclear parts.

**To Reviewer#2** Thank you for pointing out the presentation issues. Here is our revision plan.
We will rewrite Section 2 as a formal introduction to the shifting theory (or high level ideas). Since our objective is to
minimize $h \Leftrightarrow$ choosing a family of Lyapunov function that only involves $h$, a critical issue is that we cannot even
compute its gradient $\nabla h(x) = \nabla f(x) - \mu(x - x^\star)$. We figured out that in some simple cases, a change of "perspective"
is enough to access this gradient information. Take GD: $x_{k+1} = x_k - \eta\nabla f(x_k)$ as an example (which we will include
as a motivating example). We can rewrite the update as $x_{k+1} - x^\star = (1 - \eta\mu)(x_k - x^\star) - \eta\nabla h(x_k)$, and thus

$$\|x_{k+1} - x^\star\|^2 = (1 - \eta\mu)^2\|x_k - x^\star\|^2 \underbrace{-2\eta(1 - \eta\mu)\langle\nabla h(x_k), x_k - x^\star\rangle + \eta^2\|\nabla h(x_k)\|^2}_{\leq 0 \text{ if } \eta = 2/(L+\mu), \text{ which is based on the co-coercivity of } \nabla h.},$$

which is just the one-line proof of GD in the textbook (Theorem 2.1.15, [27]) but looks more structured in our opinion.
However, this change of "perspective" is too abstract for more complicated schemes. We thus encode this idea into
Lemmas 1 and 2 with some template updating rules, which serve as instantiations of the shifted gradient oracle. Then,
we can directly choose various gradient estimators for $h$ (GD, SVRG, SAGA, ...), and by applying Lemma 1, we obtain
a practical updating rule together with a classic inequality whose usage has been well studied. We will also rewrite
Lemma 1 to clarify this usage. Now we have enough building blocks to migrate existing schemes to the shifted objective
(since we can query its gradient oracle through Lemma 1). G-TM is basically NAG migrated to the shifted objective.
Technically speaking, the most important techniques in NAG (in our opinion) are Lemma 3 for $f$ and the standard
mirror descent lemma. G-TM was derived by having a shifted version of Lemma 3 for $h$ and the shifted mirror descent
lemma. We can also regard G-TM as a more "aggressive" parameter setting of NAG to get some insight, just like GD
with $\frac{2}{L+\mu}$ and $\frac{1}{L}$ learning rates. BS-SVRG/SAGA were derived by "replacing" the shifted gradient in G-TM as shifted
SVRG/SAGA estimator (through Lemma 1). We will add descriptions in each section to make their derivations clearer.

**To Reviewer#3** Thank you for your appreciation of our work. We will try to make the experiments more comprehensive.
**To Reviewer#4** Thank you for your comments and suggestions. We will improve our paper following your suggestions,
and also hope that the revision suggested by Reviewer#2 may make our ideas clearer.

**References:** (Kim and Fessler, 2016) Optimized first-order methods for smooth convex minimization. In *Math. Program.*.
(Moulines and Bach, 2011) Non-asymptotic analysis of stochastic approximation algorithms for machine learning. In *NeurIPS*.


[Meta-Review · NeurIPS 2020]

Dear authors, Thank you for submitting your clear and well-written paper. I am pleased to report that all reviewers liked your paper and see a potential for the field. When producing the final camera-ready paper, please check the reviewer's remarks to make it stronger. Thank you